# Epithelial polarization in 3D matrix requires DDR1 signaling to regulate actomyosin contractility

Pia Pernille Søgaard[1], Noriko Ito[1], Nanami Sato[2], Yasuyuki Fujita[2], Karl Matter[3], Yoshifumi Itoh[1]

**Epithelial cells form sheets and tubules in various epithelial organs and establish apicobasal polarity and asymmetric vesicle transport to provide functionality in these structures. However, the molecular mechanisms that allow epithelial cells to establish polarity are not clearly understood. Here, we present evidence that the kinase activity of the receptor tyrosine kinase for collagen, discoidin domain receptor 1 (DDR1), is required for efficient establishment of epithelial polarity, proper asymmetric protein secretion, and execution of morphogenic programs. Lack of DDR1 protein or inhibition of DDR1 kinase activity disturbed tubulogenesis, cystogenesis, and the establishment of epithelial polarity and caused defects in the polarized localization of membrane-type 1 matrix metalloproteinase (MT1-MMP), GP135, primary cilia, laminin, and the Golgi apparatus. Disturbed epithelial polarity and cystogenesis upon DDR1 inhibition was caused by excess ROCK (rho-associated, coiled-coil-containing protein kinase)-driven actomyosin contractility, and pharmacological inhibition of ROCK was sufficient to correct these defects. Our data indicate that a DDR1-ROCK signaling axis is essential for the efficient establishment of epithelial polarity.**

## Introduction

Epithelial tubules form important functional units in various epithelial organs and are composed of polarized epithelial cells. Polarized epithelial cells establish polarity and divide the plasma membrane into apical, lateral, and basal membrane domains, allowing various molecules to be secreted to specific areas of the plasma membrane. This ensures that components of the basal lamina, such as laminin and type IV collagen are secreted to the basal membrane domain, whereas other proteins, such as milk proteins in the mammary gland, are secreted at the apical surface into the lumen of the tubule. Correct orientation of polarity is, thus, essential for the functionality of epithelial organs, and establishment of apicobasal polarity is a critical step during formation of epithelial tubules.

Tubulogenesis results from coordination of fate determination of tip cells and follower cells, cell proliferation, cell adhesion to the ECM, ECM degradation, and cytoskeletal reorganization within the 3D environment. This coordination relies on epithelial polarity being established and maintained to achieve proper placement of functional molecules in the right area of the plasma membrane at the right time. Membrane-type 1 matrix metalloproteinases (MT1-MMP), a membrane-bound collagen degrading enzyme (Holmbeck et al, 2004; Itoh, 2015), is required for ECM degradation during tubulogenesis and is an example of a molecule that is regulated according to epithelial polarity (Weaver et al, 2014). Cells at the tip of forming tubules need to degrade the ECM to extend into the surrounding 3D collagen matrix. To achieve this, the cells must localize MT1-MMP at the basal side of the membrane to bring it into contact with its substrate while cells at the base of the growing tubule restrict access of MT1-MMP to the ECM by localizing it exclusively at the apical luminal surface (Weaver et al, 2014). However, the underlying molecular mechanism that drives this localization switch is unknown.

Cell–ECM interactions are important for orientation of apicobasal polarity, and ECM receptors such as integrins play important roles during polarization (Rodriguez-Boulan & Macara, 2014). A collagen receptor tyrosine kinase, discoidin domain receptor 1 (DDR1), is highly expressed in epithelial cells where it is reported to affect several cellular processes including differentiation and migration (Shrivastava et al, 1997; Vogel et al, 1997; Leitinger, 2014). DDR1 has been shown to localize at adherens junctions through association with E-cadherin, and this interaction appears to regulate DDR1 activation when cells are cultured on a collagen matrix (Wang et al, 2009). DDR1, on the other hand, stabilizes E-cadherin at the cell surface by preventing its endocytosis via inhibition of β1 integrin–mediated Src activation (Yeh et al, 2011). DDR1 has also been shown to interact with Par3/Par6 at cell–cell contacts in A431 squamous cell carcinoma cell line (Hidalgo-Carcedo et al, 2011). This interaction was shown to be essential for epithelial cancer cells to collectively migrate into a 3D matrix (Hidalgo-Carcedo et al, 2011). In contrast, a DDR1-Par3 axis has been suggested to suppress 3D invasion of the pancreatic ductal adenocarcinoma cell line CD18 (Chow et al, 2016). Despite Par3 being a central player in epithelial

[1]Kennedy Institute of Rheumatology, University of Oxford, Oxford, UK   [2]Institute for Genetic Medicine, Division of Molecular Oncology, Hokkaido University, Sapporo, Japan   [3]UCL Institute of Ophthalmology, University College London, London, UK

Correspondence: yoshi.itoh@kennedy.ox.ac.uk

polarity, the role of DDR1 in establishment of apicobasal polarity has not been examined.

Here, we show that regulation of the apicobasal distribution of MT1-MMP requires DDR1-mediated collagen signaling. Interestingly, depletion of DDR1 or pharmacological inhibition of DDR1 kinase activity not only disturbs MT1-MMP localization but also polarity of epithelial cells in a 3D collagen matrix. Selective inhibition of DDR1 kinase resulted in the formation of large cell aggregates instead of tubules or cysts, because of increased RhoA/ROCK (rho-associated, coiled-coil-containing protein kinase)-driven actomyosin contractility. These in vitro observations upon DDR1 inhibition reflect the phenotype of aberrant mammary gland branching morphogenesis in DDR1-null mice. Taken together, these results reveal a novel role for DDR1 kinase during epithelial polarization, which supports the epithelial tubulogenic programme.

# Results

### Attachment to collagen I is essential for hepatocyte growth factor (HGF)–induced basal localization of MT1-MMP

When epithelial cells undergo morphogenesis in a 3D collagen matrix, MT1-MMP, a membrane-bound collagenolytic MMP, plays an essential role (Kadono et al, 1998; Hotary et al, 2000; Weaver et al, 2014). Previously, we have found that the default localization of MT1-MMP in polarized MDCK epithelial cells is at the apical surface, whereas cells under stimulation with HGF localize MT1-MMP to the basal side where it facilitates invasion (Weaver et al, 2014). To confirm that HGF-induced changes in the apicobasal localization of MT1-MMP occur in live cells, we created MT1-MMP fused with a pH-sensitive GFP, superecliptic pHluorin (Miesenbock et al, 1998), in the ectodomain and mRFP-1 in the cytoplasmic domain (MT1F-pHluorin-RFP, Fig 1A, left). Superecliptic pHluorin

fluoresces brightly at neutral pH, whereas its emission is almost negligible at pH lower than six (Miesenbock et al, 1998). Thus, MT1F-pHluorin-RFP fluoresces in green upon appearance on the cell surface while the signal is diminished in the secretory/endocytic vesicles because pH in these compartments is around 5.5. The mRFP-1 in the cytoplasmic domain enables detection of total MT1-MMP regardless of its localization. MDCK cells stably expressing MT1-pHluorin-RFP (Fig S1A) were subjected to a tubulogenesis assay in 3D collagen gels for 4 d. As shown in Fig 1A (right), the GFP signal of MT1-pHluorin-RFP was detected only at the apical luminal surface of cells cultured without HGF treatment (Control). Upon HGF treatment, GFP signals at the basal surface were detected in cells at the tip of protruding tubules (Fig 1A, right, arrowheads). These results confirm previous observations in cells cultured in 2D on collagen films where HGF similarly induced basal localization of MT1-MMP (Weaver et al, 2014). To address the importance of cell–substratum interactions for this response, we compared the effect of HGF on cells cultured in uncoated wells (plastic) or on thin films of gelatin or collagen I. Interestingly, HGF only induced a change in the apicobasal distribution of MT1-MMP when the cells were cultured on collagen, whereas no changes in MT1-MMP localization were observed in cells cultured on plastic or gelatin (Fig 1B and C). These data suggest that specific signals from collagen are essential for epithelial cells to localize MT1-MMP at the basal surface upon HGF treatment.

### DDR1 is required for correct apicobasal localization of MT1-MMP, epithelial polarization, and lumen formation in 3D culture

The major receptors that transmit signals from collagen in epithelial cells are the collagen-binding integrins and the collagen receptor tyrosine kinase, DDR1 (Leitinger, 2011). DDR1 is activated by collagen types I, II, III, and IV but not by their denatured forms gelatin (Carafoli & Hohenester, 2013). Thus, DDR1 can transmit

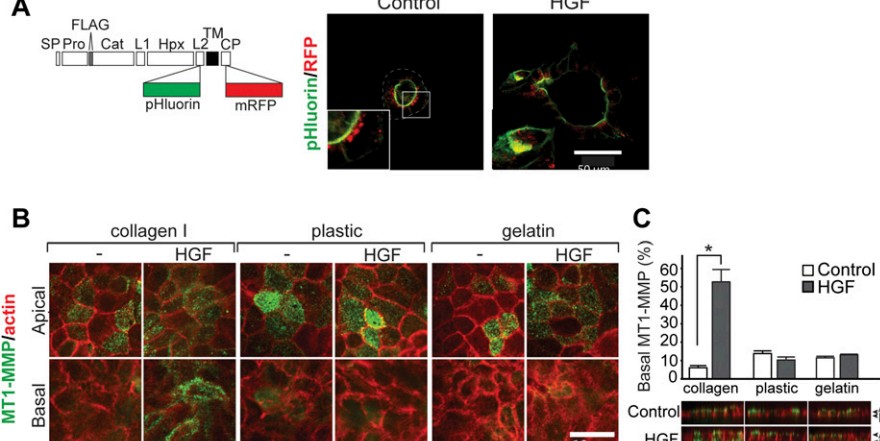

**Figure 1. HGF induced a collagen I–dependent, functionally significant increase in MT1-MMP at the basal surface of polarized, epithelial cells.**
**(A)** Left: Representation of MT1F-pHluorin-RFP construct. Right: MDCK cells stably expressing MT1-pHluorin-RFP were seeded in 3D collagen I (2 mg/ml) and cultured for 48 h without HGF (control). Culture was then continued for 20 h in the presence of HGF (50 ng/ml) (HGF). Cells were imaged with confocal microscopy for pHluorin (green) and RFP1 (red). Arrows point to MT1-MMP at the basal cell membrane. Representative images are shown. **(B)** MDCK cells were seeded on collagen I fibril, gelatin, or plastic. The cells were infected with Ad-MT1F at an MOI of five in the presence of GM6001 (10 $\mu$M). After 24 h, GM6001 was washed out and cells incubated a further 24 h in the presence or absence of HGF (50 ng/ml). Surface MT1F was stained in live cells on ice and imaged by confocal microscopy. MT1F is shown in green, actin in red. Confocal sections at the apical and basal surface are shown. Note that basal localization of MT1F can be observed upon HGF treatment when cells are cultured on collagen I. **(C)** Results from B were quantified by measuring the intensity of the 488-nm channel in identical volumes at the apical and basal surface in 10 confocal stacks per condition. Volumes were selected based on the signal in the channel representing actin. The percentage of total MT1F at the basal surface was calculated as described in the Materials and Methods section. Data are plotted as the mean ± SD. The tested conditions were compared with a Kruskal–Wallis rank sum test ($P < 2 \times 10^{-16}$) before localizing differences with a pairwise Wilcoxon test with Bonferroni–Holm's correction of the $P$-value to correct for multiple testing. The asterisks depict statistically significant differences (*$P < 0.05$). Col I: collagen I.

collagen-specific signals to the cells. In contrast, integrin-dependent adhesions can occur both on collagens and gelatin, and integrins may, therefore, not be able to provide collagen-specific signals. Moreover, a recent report has indicated that collagen-binding integrins can bind to monomeric collagen but fail to bind a fibrillar form of collagen (Woltersdorf et al, 2017). Therefore, we investigated a potential role of DDR1 in regulating the apicobasal distribution of MT1-MMP during tubulogenesis. First, we examined the effect of DDR1 knockdown on tubulogenesis by MDCK cells in a 3D collagen matrix. As shown in Fig 2A (bright field, upper panel), whereas MDCK cells expressing non-targeting shRNA (NT-shRNA) underwent tubular morphogenesis after 72 h, cells stably knocked down for DDR1 (Fig S2A) formed cellular aggregates (Fig 2A, bright field, lower panel). The organization of these aggregates was further investigated by staining for GP135, a well-recognized apical marker for MDCK cells (Ojakian & Schwimmer, 1988; Meder et al, 2005). In control tubules, GP135 was exclusively localized at the apical cell surface along the central lumen (Fig 2A, top right panel). In contrast, the cell aggregates formed by shDDR1-expressing cells failed to establish a central lumen, and GP135 distribution was disrupted and localized to the cell-ECM interface in many parts of the aggregates (Fig 2A, bottom right panel, arrowheads). Similar phenotypes were observed when DDR1 function was blocked by stable overexpression of a dominant-negative form of DDR1, a cytoplasmic domain deletion mutant (DDR1ΔC) (Wang et al, 2005) (Figs 2B and S2B), or by pharmacological inhibition of DDR1 using a selective DDR inhibitor, DDR1-IN-1 (1 µM) (Kim et al, 2013) (Fig 2C). Another apical marker is the primary cilium. This is a structure associated with the apical membrane of confluent, fully differentiated epithelial cells that can be detected by antibodies against acetylated tubulin (Torkko et al, 2008; Vieira et al, 2006). In control cells, the primary cilia were found in the apical membrane, extending towards the lumen (Fig 2D, left panel, arrow and inset), whereas only a diffused staining pattern of acetylated tubulin was found in the DDR1-IN-1–treated group (Fig 2D, right panel). Staining for laminin demonstrated that control cells deposit a laminin-containing basement membrane along their basal cell surface. In contrast, a diffused distribution of laminin was observed in DDR1-IN-1–treated cells (Fig 2D, right). We confirmed that 1 µM DDR1-IN-1 inhibited collagen-induced phosphorylation of DDR1 (Fig S2C) and that no toxicity is observed at doses up to 5 µM in a number of cell lines, including MDCK cells (data not shown) (Majkowska et al, 2017). These findings suggest a role for DDR1 in establishing or maintaining polarity during 3D tubulogenesis.

HGF-induced tubulogenesis of MDCK cells is thought to occur through induction of partial EMT in cells in polarized cysts. These partially differentiated cells then proliferate and invade as a continuous chain of cells, which through proliferation and de novo lumen formation becomes a tubular structure (Pollack et al, 1998; O'Brien et al, 2002). In a tubulogenesis assay originating from single cells, failure to form tubular structures could stem from effects on the initial formation of groups of cells or from effects on the initiation and execution of tube formation. To focus on the latter aspect, we set up an alternative tubulogenesis assay using gelatin-coated microcarrier beads (Palmisano & Itoh, 2010). In this assay system, MDCK cells are cultured to form a confluent monolayer on

the bead before culture in a 3D collagen matrix. Thus, all structures in this assay originate from a similar number of cells because of the consistent size of the beads (~175 µm diameter). MDCK cells stably transfected with mock vector, wild-type DDR1, or DDR1ΔC were cultured to confluence on microcarrier beads and suspended within a 3D collagen gel in the presence of HGF. After 72 h, Mock cells had undergone clear morphological changes: in addition to the layer in contact with the gelatin-coated bead, a second layer of cells formed, and large tubular structures protruded from this outer layer in the basal direction (Fig 2E, top middle panel, arrowheads and Fig S3). Interestingly, DDR1-expressing cells did not form tubular structures, but the lumen between the inner and outer cell layer expanded, leading to formation of large cysts around the beads (Fig 2E, center middle, arrow, also see Fig S3). The outer cell layer appeared smooth with fewer and shorter protrusions compared with the Mock cells, suggesting that DDR1 overexpression caused uniform matrix degradation and invasion along the basal surface of the outer cell layer in contrast to the spatially restricted invasion occurring only at the tip of protrusions in Mock cells. DDR1ΔC-expressing cells showed very limited invasion and tubulogenesis (Fig 2E). Furthermore, DDR1-IN-1 treatment converted both Mock and DDR1-expressing cells to a phenotype similar to the cells expressing DDR1ΔC (Fig 2E). No tubulogenesis or invasion was observed for either condition in the absence of HGF (Fig 2E). Quantification of these results confirmed a significant decrease in the length of tubular structures upon overexpression of DDR1 or DDR1ΔC (Fig 2E, top graph), whereas DDR1 overexpression caused a significant increase in the area of the structures (Fig 2E, bottom graph). These results indicate a potential pro-invasive role for DDR1 in epithelial cells. Structures formed on the beads by Mock and DDR1-expressing cells displayed a typical polarized epithelial phenotype with two layers of cells separated by a lumen highlighted by GP135 staining (Fig 2F, arrows, Fig S3). In contrast, DDR1ΔC cells grew in multiple layers on top of each other without clearly defined apical or basal sides with several microlumens (Fig 2F, arrowheads, Fig S3). This effect of DDR1 inhibition was not restricted to MDCK cells as a similar phenotype was observed in the human mammary epithelial cell line MCF10A. As shown in Fig 2G, MCF10A cells formed tubular structures on microcarrier beads in the presence of HGF, and DDR1-IN-1 treatment caused a multilayered phenotype. Thus, DDR1 appears to be required for epithelial cells to establish polarity during HGF-induced tubulogenesis.

Because regulation of the apicobasal localization of MT1-MMP is essential for tubulogenesis (Weaver et al, 2014) and collagen as a substratum is necessary for its basal localization; we next examined if DDR1-mediated collagen signaling plays a role in the basal localization of MT1-MMP in MDCK cells. As shown in Fig 2H, upon selective inhibition of DDR1, around 50% of MT1-MMP on the cell surface was found to be at the basal surface regardless of HGF treatment. In contrast, when cells were cultured on plastic or gelatin, neither DDR1-IN-1 nor HGF treatment caused a change in the localization of MT1-MMP, which remained apical (Fig 2H). These observations were corroborated in MDCK cells stably expressing MT1F-pHluorin-RFP (Fig S4). These data indicate that regardless of the presence of DDR1 signaling, a collagen substratum is required for MT1-MMP to localize to the basal membrane, but also further suggest that the presence of collagen introduces a requirement for

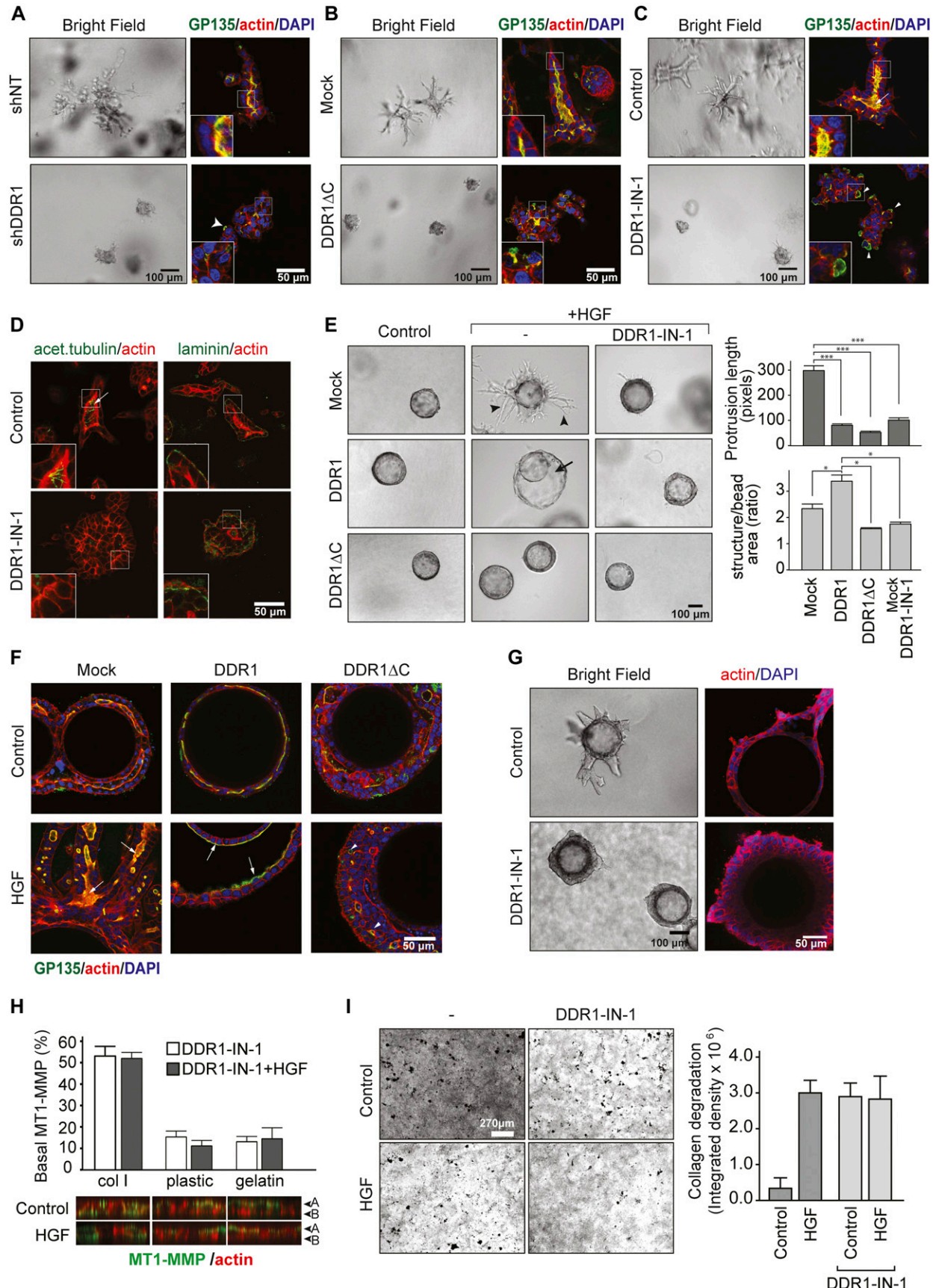

DDR1 signaling to ensure the apical localization of MT1-MMP in non–HGF-treated cells.

We confirmed that the basally localized MT1-MMP upon DDR1-IN-1 treatment is biologically active as it caused enhanced collagen degradation even in the absence of HGF treatment (Fig 2I). Because enhanced MT1-MMP–mediated collagen degradation would not be expected to inhibit invasion during tubulogenesis, mis-localization of MT1-MMP is not likely to cause the observed effect of DDR1 inhibition on tubulogenesis.

In the absence of HGF, the structures formed by DDR1-expressing cells were similar to Mock cells; however, DDR1ΔC-expressing cells still displayed a multilayered phenotype (Figs 2E and F, and S3), suggesting that the lack of epithelial organization upon inhibition of DDR1 is not related to HGF signaling. To explore this possibility further, we analyzed the effect of DDR1 inhibition on cyst formation by MDCK cells in 3D collagen matrices in the absence of HGF. As shown in Fig 3A, control cells formed cysts consisting of a mono-layer of cells with a well-organized central lumen highlighted by GP135 staining and a typical distribution of laminin along the basal surface (Fig 3A, top panels). It is known that the Golgi apparatus localizes to the apical side of the nucleus in a manner depending on the polarized organization of the microtubule network (Barr & Egerer, 2005). In control cells, the cis-Golgi resident protein GM130 localized in the apical part of the cytoplasm towards the cyst lumen as expected (Fig 3A, top right panels). In contrast, cells treated with DDR1-IN-1 formed cellular aggregates without a central lumen and with disorganized staining patterns of GP135, laminin and markers of the primary cilia and cis-Golgi (Fig 3A, bottom panels), confirming that apicobasal polarity was disturbed. These observations were also confirmed in DDR1ΔC-expressing cells, which similarly formed structures that lacked a central lumen and primary cilia and exhibited aberrant GP135 localization (Fig S5). The effect of DDR1 inhibition on cystogenesis was validated in CaCO-2 cells, a well-differentiated human colorectal carcinoma cell line. As shown in Fig 3B, whereas culturing CaCO-2 cells within 3D Matrigel resulted in

formation of polarized cysts (upper panels), DDR1-IN-1 treatment caused the formation of cellular aggregates with no central lumen (lower panels). Tight junctions were formed towards the apical surface in the control cells as expected, whereas they were present at the interface between the cells and the collagen matrix in the DDR1-IN-1–treated cells (Fig 3B, lower left panel). The Golgi marker GM130 was also diffusely distributed in the aggregates, compared with its localization in the apical part of the cytoplasm in control cysts (Fig 3B, right panels). The role of DDR1 in formation of epithelial polarity, thus, appears to be a common feature of different epithelial cell lines. Taken together, we conclude that DDR1 is a fundamental signaling molecule that is essential for establishment of epithelial polarity required for proper MT1-MMP localization on the cell surface, tubulogenesis, and lumen formation.

## Inhibition of DDR1 delays maturation of epithelial monolayers

In contrast to 3D culture, inhibition of DDR1 in 2D culture did not affect the localization of the apical marker GP135, the tight junction marker ZO-1, or E-cadherin (Fig 4A). These findings were also confirmed in DDR1ΔC-expressing cells (Fig S6A). To further examine the role of DDR1 in formation of functional epithelial layer, the trans-epithelial resistance (TER) was measured during the maturation of the monolayer. MDCK cells were initially seeded onto collagen-coated culture inserts in a $Ca^{2+}$-free medium. Junction formation was then initiated by switching to a $Ca^{2+}$-containing medium, and TER was measured between 0 and 25.5 h (Fig 4B, a) and between 23 and 69 h (Fig 4B, b). In control cells, TER values rose within the first 8 h during the formation of epithelial junctions (Fig 4B, a, black line). Between 20 and 25 h, TER decreased rapidly and then continued to decrease until it reached a homeostatic level (Fig 4B, b, black line). This decrease in TER is likely due to insertion of ion channels and ion-permeable claudins in the plasma membrane in a homeostatic level; thus, it indicates that polarization/maturation of the monolayer is complete. This profile of TER development is typical of low-resistance

---

**Figure 2. DDR1 kinase is required for tubulogenesis and polarity-dependent MT1-MMP localization at the basal surface.**
**(A)** MDCK cells stably expressing nontarget-shRNA (siNT) or DDR1 shRNA (siDDR1) were cultured in 3D collagen I gels in the presence of HGF (50 ng/ml) for 5 d. The cells were then imaged for bright field and/or subjected to indirect immunofluorescence imaging for GP135 (green), F-actin (red), and nuclei (DAPI/blue). Insets show enlarged images of boxed area. **(B)** MDCK cells stably transfected for empty plasmid (Mock) or cytoplasmic domain-deleted DDR1 (DDR1ΔC) were subjected to tubulogenesis assays and analyzed and imaged as that of (A). **(C)** MDCK cells were subjected to tubulogenesis assay in the presence of DMSO vehicle (control) or DDR1-IN-1 (1 µM). The cells were analyzed and imaged as that of (A). **(D)** MDCK cells were subjected to tubulogenesis assay in the presence of DMSO vehicle (control) or DDR1-IN-1 (1 µM) and imaged by indirect immunofluorescence for acetylated tubulin (green, left), laminin (green, right), and F-actin (red, left and right) as indicated. **(E)** MDCK cells stably expressing empty plasmid (Mock), DDR1, and DDRΔC were cultured on gelatin-coated microcarrier beads to the confluent. Beads were then embedded in collagen gels with or without HGF and/or DDR1-IN-1 (1 µM) as indicated. Bright-field images were captured after 72 h. Representative images are shown from 1 of >3 independent experiments. Arrowheads point to tubular structures generated. Arrows point to the lumen formed between the cells in contact with the bead and the additional outer layer of cells. In the right top graph, quantification of the length of the 10 longest protrusions of five different structures per condition measured using ImageJ software are shown. The data are plotted as the mean ± SE. Kruskal test revealed a significant difference on the distribution of the groups ($P < 2.2 \times 10^{-16}$), and differences were localized with a pairwise Wilcoxon test with Bonferroni–Holm's correction for multiple testing. The asterisks depict statistically significant differences (\*\*\*$P < 0.001$). The bottom graph indicates quantification of the area of structures: the areas of 5–7 structures per condition were measured using ImageJ software. To correct for differences in bead size and thereby different number of cells at the starting point, the area of each structure was normalized to the area of the bead. The data are plotted as the mean ± SE. Kruskal test revealed a significant difference on the distribution of the groups ($P = 0.0004$), and the differences were localized with a pairwise Wilcoxon test with Holm correction for multiple testing. The asterisks depict statistically significant differences (\*$P < 0.05$). **(F)** Cells from (D) were subjected to indirect immunofluorescent staining for GP135 (green), actin (red), and nuclei (DAPI, blue). Arrows point to the apical lumen. Note that cells expressing DDR1ΔC do not form lumen and disturbed for GP135 localization. **(G)** MCF10A cells were cultured as of (E) in the presence of DMSO (Control) or 1 µM DDR1-IN-1 for 7 d. Gels were fixed and imaged for bright field or subjected to staining for F-actin (red) and DAPI (blue) and analyzed by confocal microscopy. **(H)** MDCK cells were seeded on collagen I fibril, plastic, or gelatin film in the presence of DDR1-IN-1 (1 µM) for 24 h. The cells were then transduced with Ad-MT1F for 24 h and further cultured for 16 h in the presence or absence of HGF (50 ng/ml) with DDR1-IN-1. % of basal surface–localized MT1-MMP was analyzed as of Fig 1C. Representative cross sections are shown. The data are plotted as the mean ± SD. **(I)** MDCK cells were subjected to collagen film degradation assay in the presence or absence of HGF and/or DDR1-IN-1. Digested areas were quantified and shown in the graph (n = 4).

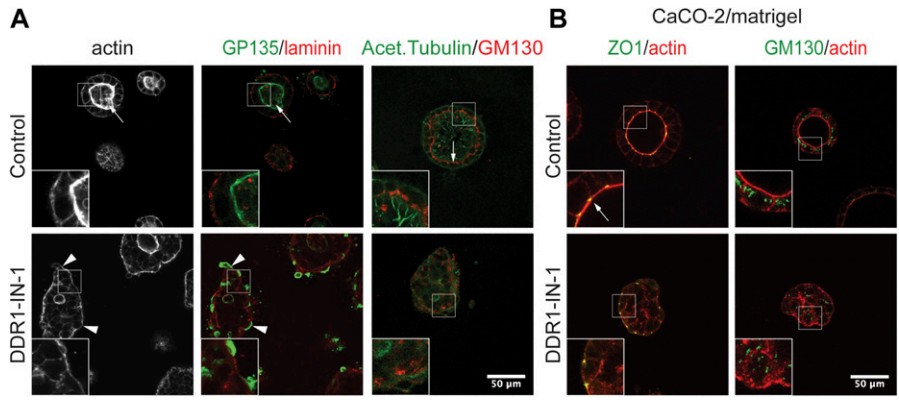

**Figure 3. Inhibition of DDR1 prevents formation of polarized cysts in 3D.**
**(A)** Left and middle images: MDCK cells were grown in the 3D collagen gel in the presence of DMSO (control) or 1 µM DDR1-IN-1 for 7 d and subjected to indirect fluorescent imaging for F-actin (grey), GP135 (green), and laminin (red). Right images: cells were also were stained for acetylated tubulin (green) and GM130 (red). Arrows point to the lumen of cysts, whereas arrowheads point to basal staining of GP135 in the presence of DDR1-IN-1. **(B)** CaCO-2 cells were cultured in 2% Matrigel in the presence of DMSO (control) or DDR-IN-1 (1 µM) for 9 d. Left: cells were stained for ZO-1 (green) and f-actin (red). Right: cells were stained for GM130 (green) and f-actin (red). Representative images are shown.

MDCK cells (Gonzalez-Mariscal et al, 1990). DDR1-IN-1 treatment caused a slight lag in the initial rise in TER between 6 and 8 h after the Ca2+ switch (Fig 4B, a), whereas TER at 20–30 h was notably higher than that of the control cells (Fig 4B, a and b, red line), indicating a delay in the formation of epithelial polarity. After 69 h, control and DDR1-IN-1–treated cells had similar TER values, suggesting that junctions with full barrier function seem to have eventually formed in the absence of DDR1 signaling (Fig 4B, b). Interestingly, DDR1-IN-1 also affected the TER of cells on plastic (Fig S6B), suggesting that the function of DDR1, in this case, does not depend on collagen recognition at the basal cell surface. These data indicate that DDR1 inhibition causes a delay in the formation of a fully matured epithelial monolayer under 2D culture conditions. In contrast to the situation in 3D culture, the effects of DDR1 inhibition resolved over time, indicating that studies to address the role of DDR1 in formation of polarity in 2D must be carried out at sufficiently early time points.

### DDR1 inhibition alters cytoskeletal organization through increased junctional MLC activity

During the course of the experiments, we found that DDR1 inhibition affected cell shape, which is consistent with previous studies implicating DDR1 in regulation of the cytoskeleton (Huang et al, 2009; Yeh et al, 2009; Hidalgo-Carcedo et al, 2011; Rhys et al, 2018). Because cytoskeletal dynamics are essential during the establishment of epithelial polarity, we next investigated if the effect of DDR1 inhibition on epithelial polarization is related to cytoskeletal organization. Because the delay in the formation of a mature monolayer upon DDR1-IN-1 treatment was most clear between 24 and 48 h (Fig 4B), we chose to examine cytoskeletal organization 36 h after seeding the cells. As shown in Fig 4C, control MDCK cells on collagen formed islands of cells with a collectively organized cytoskeleton, comprising a thick cortical actin band along the periphery of the islands, which was positive for diphosphorylated myosin light chain (ppMLC) (Fig 4C, left panels, arrows). Actin bands along cell–cell contacts within the cell island were weak and showed negligible ppMLC staining (Fig 6A, left panels). In contrast, DDR1-IN-1–treated cells formed thicker actin bundles along cell–cell contacts, which were positive for ppMLC (Fig 4C, middle panels, arrows). Inhibition of ROCK, which is the major promotor of MLC phosphorylation, by Y27632 caused the ppMLC signal at the cell–cell

junctions to disappear (Fig 4C, right panels). This suggests that DDR1 may play a role in the spatial regulation of MLC activity, potentially through modification of RhoA-ROCK signaling. Interestingly, a similar increase in ppMLC along the cell–cell junctions was seen upon DDR1 inhibition in cells cultured on glass (Fig S7A). This suggests that the effect of DDR1 on phosphorylation of MLC does not depend on the recognition of a collagen substratum, similar to the effect on TER levels (Figs 4D and S6B).

We further analyzed the localization of ppMLC along with ZO-1 in confluent MDCK cells. Compared with control cells, DDR1-IN-1–treated cells showed an increase in ppMLC along cell–cell contacts in confocal sections at the apical cell surface (Fig 4D, bottom, left, arrows point ppMLC-positive cell–cell contacts). An overall shift in the distribution of ppMLC along the apicobasal axis was seen when the mean intensity in each confocal section was normalized to the highest intensity in the stack and plotted against the Z-axis (Fig 4E). In control cells, the highest mean intensities were seen in sections at the basal surface, whereas treatment with DDR1-IN-1 shifted the highest ppMLC intensity towards the apical cell surface (Fig 4E, graph). Inhibition of DDR1, thus, causes an increase in ppMLC along the apical part of the lateral membrane domain. Interestingly, the majority of DDR1 was found to localize in the lateral cell membrane where it co-distributed with E-cadherin (Fig 4F). ZO-1 localized just apical side of DDR1 with little colocalization observed (Fig S7B). DDR1 may, thus, contribute to the coordination of cytoskeletal organization between neighbouring cells through local inhibition of ROCK-dependent phosphorylation of MLC in the apical part of the lateral plasma membrane.

### Suppression of ROCK signaling by DDR1 is required for apicobasal polarization and lumen formation during cystogenesis

Spatiotemporal regulation of RhoA-ROCK activity is essential during polarization and morphogenesis, and inappropriate activation can interrupt these processes. Indeed, inhibition of RhoA-ROCK signaling has been found to rescue defective cystogenesis in MDCK cells where polarity had been disrupted by inhibiting the function of integrin β1 (Yu et al, 2008; Myllymaki et al, 2011; Bryant et al, 2014), integrin α2 (Myllymaki et al, 2011), PKCαβ (Bryant et al, 2014), or ARF6 (Monteleon et al, 2012). We, therefore, next asked if the aberrant, local increase in ROCK activity observed upon inhibition of DDR1 is

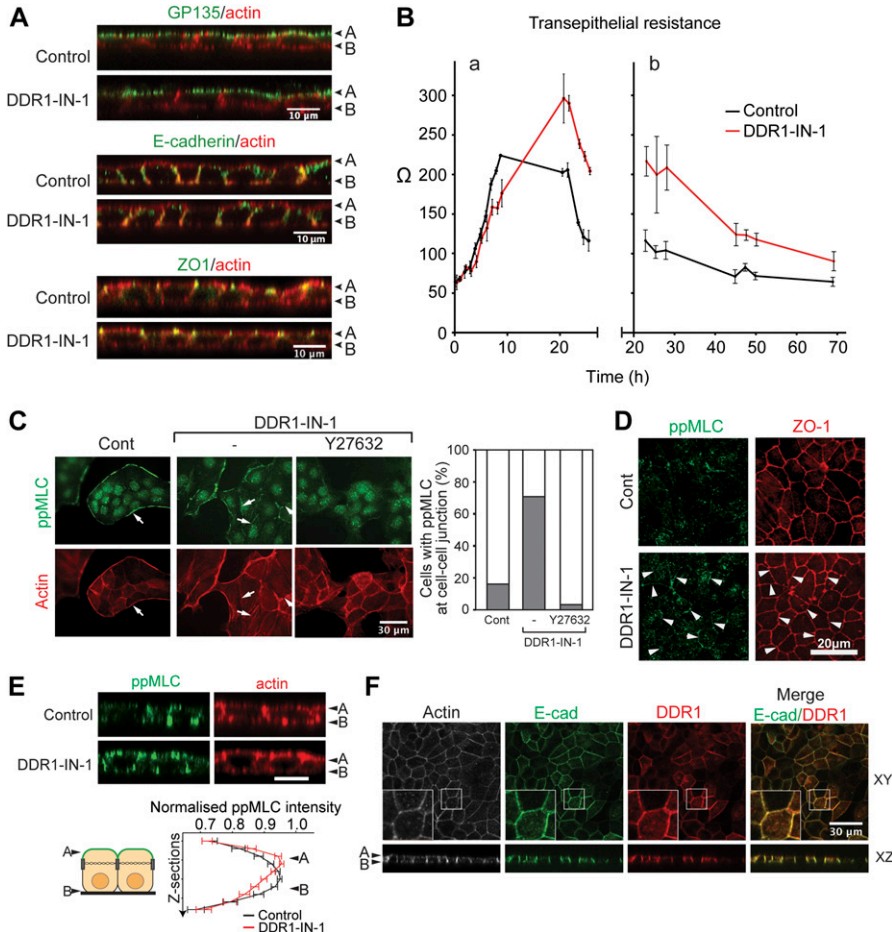

**Figure 4.   DDR1 inhibition result in delayed establishment of apicobasal polarity and ROCK-dependent ppMLC accumulation at cell–cell junction at the apical surface.**
**(A)** MDCK cells were cultured on collagen I at 100% confluency in culture inserts in the presence or absence of DDR1-IN-1 (1 µM) for 4 d and subjected to indirect immune fluorescent staining for GP135 (green) and F-actin (red) (top), E-cadherin (green) and F-actin (red) (middle); and ZO-1 (green) and F-actin (red) (bottom). Images were captured by confocal microscopy. Representative XZ sections are shown. **(B)** MDCK cells cultured with or without DDR1-IN-1 were subjected to TER assay as described in the Materials and Methods section. TER measurements were performed 23, 25.5, 28, 45, 47.5, 50, and 69 h after addition of calcium. The average TER values of three wells per condition were plotted against time. Error bars represent SD between the wells. **(C)** MDCK cells were seeded at 40% confluency on collagen film for 36 h in the presence or absence of DDR1-IN-1 (1 µM) and/or Y27632 (10 µM) as indicated. The cells were stained for ppMLC (green) and F-actin (red). Arrows point to the common cortical actin bands in control (Cont) cells and ppMLC-positive stress fibres in DDR1-IN-1–treated cells. Images were analyzed by calculating and plotting the percentage of cells with ppMLC-positive actin bundles along cell–cell contacts. Analysis of 122–262 cells per condition from two independent experiments is shown. **(D)** MDCK cells were cultured at 100% confluence in culture inserts coated with collagen I in the presence of DMSO or DDR1-IN-1 (1 µM) as indicated. The cells were fixed and stained against ppMLC (green) and ZO-1 (red). Arrowheads point to junctional ppMLC. **(E)** MDCK cells cultured as above were stained with ppMLC (green) and actin and imaged with confocal microscope. Top: representative XZ sections are shown for ppMLC (green) and actin (red). Scale bar, 10 µm. Bottom: the average intensity in each confocal section measured in ImageJ and normalized to the highest average intensity in that stack. Stacks were aligned based on the most apical section and the mean

at each Z section calculated and plotted with error bars representing standard error of the mean. 55–60 stacks per condition pooled from four independent experiments were analyzed. (A) apical, (B) basal. **(F)** MDCK-DDR1 cells were cultured to confluency in collagen I–coated culture inserts for 36 h. The cells were stained for: F-actin (grey), DDR1 (red), and E-cadherin (green). The samples were analyzed by confocal microscopy and representative images of XY and XZ planes shown. (A) apical, (B) basal.

relevant for the inability of these cells to polarize in 3D collagen. To investigate this, we added ROCK inhibitors to DDR1-IN-1–treated MDCK cells undergoing cystogenesis in 3D collagen. As observed earlier, inhibition of DDR1 disrupted the formation of cysts (Fig 5A and C). Interestingly, inhibition of ROCK by either Y27632 (10 µM) or H1152 (4 µM) rescued the ability of DDR1-IN-1–treated MDCK cells to form cysts with a central lumen and correct apical distribution of GP135, whereas inhibition of ROCK alone did not affect cystogenesis (Fig 5A and C). The major role of ROCK in cystogenesis appeared to be mediated via its effects on myosin activity as seen by bleb-bistatin (25 µM), a myosin II inhibitor, similarly being able to rescue cystogenesis in the absence of DDR1 signaling (Fig 5B and C). This suggests that suppression of MLC phosphorylation through suppression of ROCK signaling is an important function of DDR1 during epithelial morphogenesis.

## DDR1 is involved in branching morphogenesis and epithelial organization of mammary epithelium in vivo

During pregnancy and lactation, female DDR1-null mice display hyperproliferation and abnormal branching of the mammary

ducts, which is accompanied by impaired lactation and, thus, inability to sustain viable offspring (Faraci-Orf et al, 2006; Vogel et al, 2001). To investigate if this phenotype in vivo is related to the role of DDR1 during polarization and morphogenesis of epithelial cells in vitro, we next investigated the organization of the mammary epithelium in lactating DDR1-null C57Bl/6J females. Mammary glands were collected from three wild-type and three DDR1-null females, 3 d postpartum. HE staining revealed wild-type mammary glands to be filled with dilated alveoli at this time point (Fig 6A, top left panel). In contrast, the mammary glands of DDR1-null mice were composed of more fat tissue relative to epithelial structures. Furthermore, alveoli were noticeably smaller in the DDR1-null mice (Fig 6A, bottom left panel). Quantification of the area of alveoli in wild-type and DDR1-null mice showed that the mean size in DDR1-null mice was decreased by ~60–70% compared with wild-type animals (Fig 6A, right). This decrease in alveolar size and increase in fat tissue in DDR1-null mice agrees with previous reports on the involvement of DDR1 in the remodeling of epithelial structures in the lactating mammary gland (Vogel et al, 2001; Faraci-Orf et al, 2006). To evaluate if these changes in epithelial architecture involved an altered apicobasal

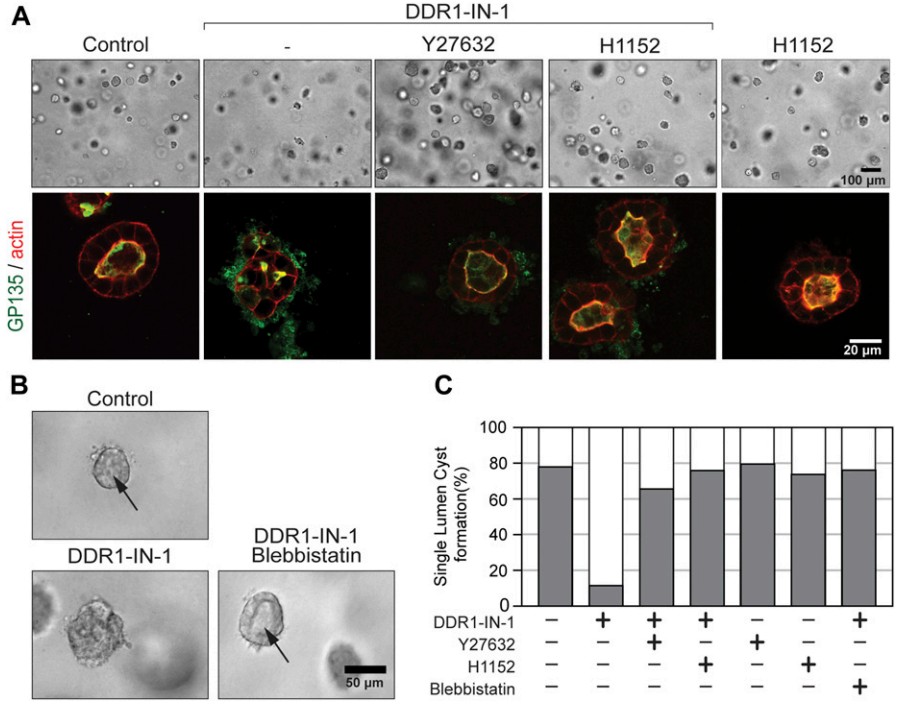

**Figure 5.  Inhibition of ROCK activity rescues formation of polarized cysts in the absence of DDR1 signaling.**
**(A)** MDCK cells were cultured in 3D collagen gel for 5 d in the presence or absence of DDR1-IN-1 (1 μM), Y27632 (10 μM), and/or H1152 (4 μM). The cells were fixed and stained for GP135 (green) and F-actin (red). The samples were analyzed by confocal microscopy. **(B)** The cells were cultured as in (A) in the presence of DDR1-IN-1 (1 μM) and/or blebbistatin (25 μM). Bright-field images were captured by a widefield microscope. Arrows point to the lumens. **(C)** Population of the cysts having monolayer lumen was counted from each treatment and percentages of the cysts within total structures are shown (grey). Total n number of structures incorporated in this analysis are 460 (control), 432 (DDR1-IN-1), 771 (DDR1-IN-1 + Y27632), 250 (DDR1-IN-1 + H1152), 231 (Y27632), 142 (H1152), and 134 (DDR-IN-1 + blebbistatin).

organization, we stained the tissue sections with antibodies against the basement membrane components collagen IV and laminin. Collagen IV localized continuously along the basal side of alveoli in the wild-type mammary glands in correspondence with the expected distribution of basement membrane components (Fig 6B, left panels). In contrast, the staining pattern in DDR1-null mice was more diffused with a punctate rather than continuous distribution along the basal cell surface and increased occurrence of intracellular staining (Fig 6B, right panels). Similar observations were made for laminin distributions in the glands (Fig 6C). This suggests that the lack of DDR1 disrupts the polarized secretion and assembly of basement membrane components during remodeling of the mammary epithelium throughout lactogenic differentiation. The role of DDR1 in vivo may, thus, involve establishment of polarity during 3D morphogenesis, similar to what was observed in in vitro experiments. In contrast to the in vitro setting, remodeling of the mammary epithelium in vivo occurs in a complex microenvironment with several cell types present. The observed phenotype could, thus, be caused by the loss of DDR1 in other cell types than epithelial cells. To investigate if there is an intrinsic requirement for DDR1 signaling in primary, mammary epithelial cells undergoing branching morphogenesis, we isolated mammary organoids from 12-to 14-wk-old C57Bl/6J mice and induced branching morphogenesis by addition of bFGF. Control organoids reorganized into round structures after overnight incubation and remained like this throughout 9 d of culture (data not shown). In contrast, bFGF-treated organoids extended tubular structures after 5 d of culture (Fig 6D). Interestingly, DDR1-IN-1 (0.5 μM) treatment reduced the percentage of organoids undergoing branching morphogenesis by 47% compared with bFGF-treated organoids (Fig 6D). This suggests that the role for

DDR1 in tubulogenesis of epithelial cell lines in vitro indeed may reflect the function of DDR1 in the mammary gland in vivo.

## ROCK suppression by DDR1 contributes to lactogenic differentiation of mammary epithelial cells

We next asked whether the observed role of DDR1 in regulating ROCK activity during morphogenesis in MDCK cells could have relevance for the impaired lactogenic differentiation in DDR1-null mice (Vogel et al, 2001). Indeed, timely down-regulation of ROCK activity has previously been associated with lactogenic differentiation of mammary epithelial cells (Wozniak et al, 2003; van Miltenburg et al, 2009; Du et al, 2012). To study the role of DDR1 in lactogenesis, we used HC11 cells, which is a clonal derivative of the COMMA-1D mouse mammary epithelial cells, selected based on their ability to differentiate and produce milk proteins in response to prolactin (Ball et al, 1988; Morrison & Cutler, 2009). Similar to the in vivo situation, lactogenic differentiation of HC11 cells has a morphogenic aspect reflected by the formation of dome-shaped 3D structures termed "mammospheres" in response to lactogenic stimuli (Ball et al, 1988; Morrison & Cutler, 2009). The formation of these mammospheres can be used as a visible readout of the degree of lactogenic differentiation by these cells (Morrison & Cutler, 2009). HC11 cells were first cultured to confluency in the presence of EGF for 5 d. EGF was then removed for 24 h to make the cells competent to respond to lactogenic differentiation media containing prolactin, insulin, and dexamethasone (DIP medium). As previously reported (Morrison & Cutler, 2009), mammospheres could be observed after 3 d with numbers increasing until the experiment was terminated at day 5 (Fig 7A). Confocal microscopy of DIP-treated HC11 cells stained for actin and DAPI confirmed that

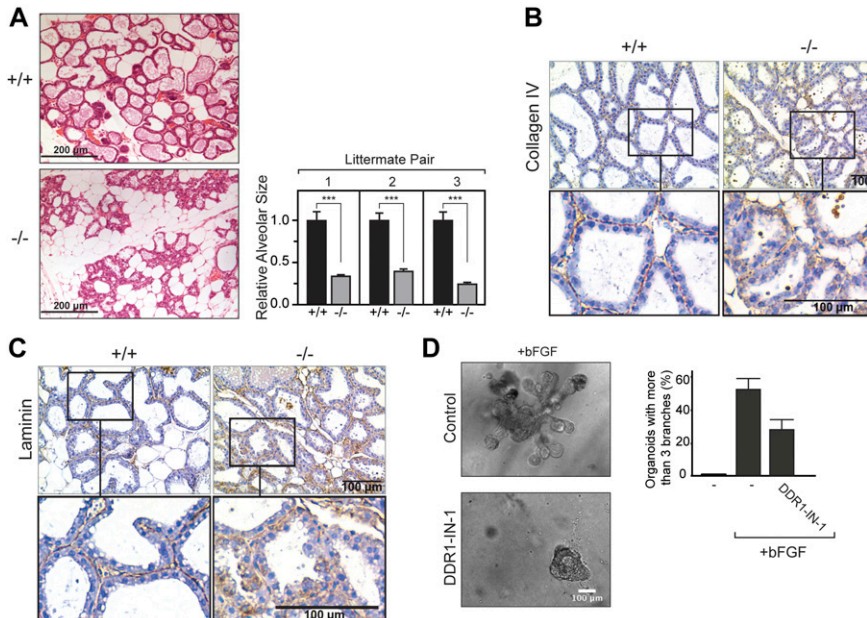

**Figure 6. DDR1 plays an intrinsic role in primary epithelial cells undergoing branching morphogenesis and is required for correct organization of the basement membrane in vivo.**
**(A)** Left: The fourth pair of mammary glands from wild-type and DDR1-null female C57Bl/6J mice were dissected out 3 d postpartum and fixed in formalin. All mice were 13–14-wk old. Tissues were embedded in paraffin and processed for HE staining. Representative tissue sections from wild-type (+/+) and DDR1-null (−/−) mice are shown. Scale bars represent 200 μm. Right: The areas of mammary alveoli were measured using ImageJ. The mean area of 99–196 alveoli per mouse was determined for three sets of wild-type (+/+) and DDR1-null (−/−) mice littermate pairs. The relative alveolar area of the mice is plotted as a bar graph for each littermate pair. The data are plotted as the mean ± SE and statistics analyzed by a t test. The asterisks depict statistically significant differences (***$P < 0.0001$). **(B, C)** Tissue sections from three wild-type and three DDR1-null mice were subjected to IHC for collagen IV (B) or laminin (C). The sections were counterstained with haematoxylin. Representative images are shown. **(D)** The second, third, and fourth pair of mammary glands were dissected from two wild-type female C57Bl/6J mice between 12 and 13 wk of age and subjected to morphogenic assay as described in the Materials and Methods section. Left: representative bright-field images are shown. Right: the percentage of organoids undergoing branching morphogenesis with more than three branches was counted. The average percentage from three independent experiments is plotted in the bar graph with error bars representing SD.

mammospheres are dome-shaped structures with a closed, hollow lumen (Fig 7B). To analyze the role of DDR1 in the morphogenic response of HC11 cells to lactogenic stimulation, we compared the number of mammospheres per field between control cells and DDR1-IN-1–treated cells. Interestingly, inhibition of DDR1 caused a significant decrease in the number of mammospheres, which was reversed by treatment with the ROCK inhibitor H1152 (Fig 7C), suggesting that excessive ROCK activity is involved in this phenotype. However, inhibition of ROCK activity on its own increased the formation of mammospheres to the level above the control cells (Fig 7C), imposing the possibility that the improvement of mammosphere formation by ROCK inhibition in DDR1-IN-1–treated cells is the sum of positive effects from ROCK inhibition and negative effects from DDR1 inhibition. However, DDR1-IN-1 treatment significantly increased the total levels of ppMLC, and this increase was counteracted by H1152 treatment (Fig 7D), suggesting that the effect of DDR1-IN-1 is mediated through the RhoA-ROCK axis. The relevance of DDR1-mediated modulation of ROCK activity for lactogenesis was further examined by evaluating the expression of β-casein in response to DIP treatment. Stimulation with the DIP medium for 4 d led to an increase in β-casein mRNA in control cells (Fig 7E). This increase was markedly reduced when HC11 cells were cultured in the presence of DDR1-IN-1, confirming that the impaired morphological response to prolactin in DDR1-inhibited cells is a readout for inefficient lactogenic differentiation. Treating the cells with H1152 alone increased the amount of β-casein mRNA to higher levels than those observed for control cells, similar to what we observed in the mammosphere formation assay (Fig 7E).

These observations confirm previous reports of DDR1 being implicated in lactogenic differentiation (Vogel et al, 2001) and

suggest that the phenotype of the mammary epithelium in DDR1-null mice may involve excessive ROCK activity.

## Discussion

Cell–matrix interactions are known to be important for the initial establishment and orientation of polarity during morphogenesis. As such, the small GTPase, Rac1, is required for the formation of polarized cysts because of its role in orchestrating laminin secretion and assembly at the basal cell surface (O'Brien et al, 2001). Similarly, the β1 integrin subunit, a component of laminin-binding integrins, is required during polarization of epithelial cells in a 3D matrix (Akhtar & Streuli, 2013; Bryant et al, 2014). In this report, we provide evidence that another matrix-recognizing receptor protein, DDR1, also plays an important role in epithelial polarity. We found the kinase activity of DDR1 to be crucial for the polarization process in 3D, since knockdown of DDR1, overexpression of a dominant-negative DDR1 mutant, and pharmacological selective inhibition of DDR1 kinase activity all resulted in the formation of non-polarized cell aggregates instead of tubules or cysts. Interestingly, it appears that the DDR1 activity involved in regulating polarity occurs not at the basal surface recognizing collagen but rather at the lateral membrane at E-cadherin–positive cell–cell junctions.

Our data suggest that an important function of DDR1 kinase activity in epithelial cells is to suppress RhoA-ROCK–dependent myosin activity along cell–cell junctions with important consequences for epithelial polarization during 3D morphogenesis. Involvement of DDR1 in regulating actomyosin contraction at adherens junctions has previously been suggested in squamous cell carcinoma A431 cells (Hidalgo-Carcedo et al, 2011). However, in

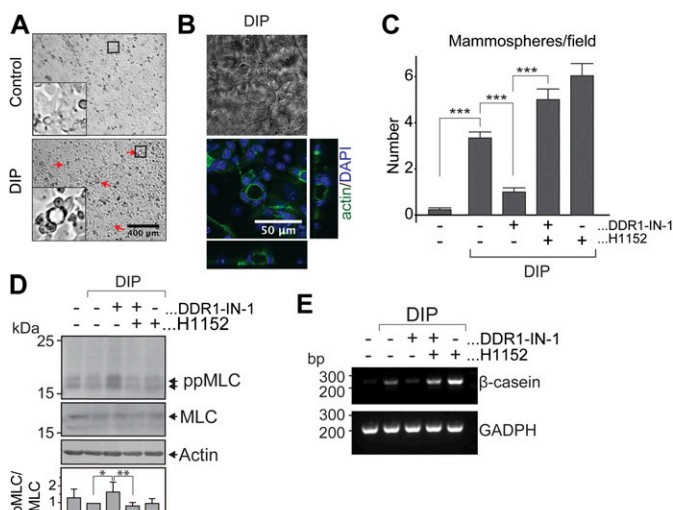

**Figure 7. Inhibition of ROCK rescues attenuated mammosphere formation and induction of β-casein expression upon inhibition of DDR1 signaling.**
**(A)** HC11 cells were subjected to lactogenic differentiation assay as described in the Materials and Methods section. Representative images of cells cultured in the normal growth media (Control) and differentiation media (DIP) from one of three independent experiments are shown. Red arrows indicate the mammospheres formed upon differentiation. **(B)** Differentiated HC11 cells were stained for F-actin (green) and DAPI (blue) and analyzed by confocal microscopy. Representative XY, XZ, and YZ sections from one experiment are shown. **(C)** HC11 were cultured as of (A) in the presence or absence of DDR1-IN-1 and/or H1152, and differentiation was quantified by calculating the mean number of mammospheres per field for each condition (10–30 fields per condition) at day 4 after adding the DIP medium. Graph represents the mean ± SD of three independent experiments and statistics analyzed by a t test. The asterisks depict statistically significant differences (***P < 0.001). **(D)** HC11 cells were cultured as of (C). 5 d after addition of the DIP medium, the cells were lysed in the presence of okadaic acid. The lysates were subjected to Western blot with primary antibodies against ppMLC, MLC, and actin. Representative blot is shown. Band intensities of ppMLC and MLC were analyzed using ImageJ and the normalized ratio between ppMLC and MLC for each treatment was shown as a graph on the bottom. The data are plotted as the mean ± SD of five independent experimental data and statistics analyzed by a t test. The asterisks depict statistically significant differences (*P < 0.03; **P < 0.015). **(E)** The cells were cultured as that of (C) and (D). Total RNA was extracted after 5 d of DIP stimulation, and RT-PCR used to evaluate the amount of β-casein and GADPH mRNA. Representative result from one of three independent experiments is shown.
Source data are available for this figure.

this system, a kinase-dead DDR1 mutant was able to rescue the increased actomyosin tension in cells silenced for DDR1 expression, demonstrating that kinase activity was not necessary to control ROCK activity (Hidalgo-Carcedo et al, 2011). As a mechanism connecting DDR1 with regulation of actomyosin contraction, they proposed that complex formation of DDR1 with Par3/Par6 at the cell–cell junctions recruits RhoE resulting in antagonization of ROCK-driven actomyosin contractility in this location (Hidalgo-Carcedo et al, 2011). We have examined whether this pathway acts in MDCK cells by attempting to co-immunoprecipitate Par3 with DDR1 and by overexpression of RFP-tagged RhoE in MDCK cells. However, our data showed that Par3 was not co-immunoprecipitated with DDR1 and RhoE localization was unchanged by DDR1 inhibition (data not shown). These results, together with our observation that DDR1-mediated ROCK suppression is kinase dependent, suggest that the mode of action of

DDR1 in our cell system differs from that in A431 cells (Hidalgo-Carcedo et al, 2011). As a molecular link between DDR1 and RhoA-ROCK in MDCK cells, Src homology-2 domain-containing phosphotyrosyl phosphatase 2 (SHP-2) may play a role. SHP-2 is activated downstream of DDR1 signaling (Wang et al, 2006) and has been reported to negatively regulate RhoA through inhibition of Vav2, a Rho guanine nucleotide exchange factor, in MDCK cells (Kodama et al, 2000). SHP-2 interacts with phosphorylated DDR1 (Wang et al, 2006) and a DDR1 phosphopeptide pull-down assay has suggested Vav2 to interact with pY484 on DDR1 (Lemeer et al, 2012), raising the possibility that DDR1 as a scaffold for both proteins could locally enhance SHP-2–mediated inhibition of Vav2. It is, thus, possible that SHP-2 activated downstream of DDR1 signaling at the lateral membrane suppresses RhoA/ROCK signaling through inhibition of Vav2. Inhibition of DDR1 would, therefore, result in an increase in Rho/ROCK activity along cell–cell contacts.

Our data indicated that the major localization of DDR1 in polarized epithelial cells is in the lateral plasma membrane just below the tight junctions (Fig 4F). This lateral localization of DDR1 has been shown to be due to its association with E-cadherin (Wang et al, 2009; Hidalgo-Carcedo et al, 2011; Yeh et al, 2011; Chen et al, 2016a; Rhys et al, 2018). E-cadherin may, thus, act as scaffolding for DDR1 to function in polarized epithelial cells. However, it is not clear how DDR1 becomes activated at the junctions. We observed that DDR1 kinase inhibition in MDCK cells cultured on plastic also resulted in increased ppMLC levels at cell–cell contacts, suggesting that the kinase activity of DDR1 at cell–cell junctions is collagen substratum independent. We detected low levels of constitutively phosphorylated DDR1 in confluent MDCK cells without collagen stimulation, and this phosphorylation was sensitive to DDR1-IN-1 treatment (Fig S2C). It is, thus, possible that a phosphorylated form of DDR1 exists at the cell–cell junction where it decreases myosin activity through suppression of the RhoA/ROCK pathway. It is known for other RTKs that overexpression can cause auto-phosphorylation of the receptor even in the absence of ligands, and clustering of DDRs has recently been shown to be implicated in their activation (Juskaite et al, 2017). It is, thus, possible that a highly clustered localization of DDR1 at the cell–cell junction may be enough to cause auto-phosphorylation of DDR1.

The default secretory pathway of MT1-MMP in polarized epithelial cells is towards the apical surface, but upon morphogenic stimulation, a population of cells destined to become tip cells begin to localize MT1-MMP at the basal surface (Fig 1) (Weaver et al, 2014). We found this HGF-induced basal localization to be collagen attachment dependent. Interestingly, DDR1 signaling was required to prevent basal localization of MT1-MMP in the absence of morphogen when cells were cultured on collagen, whereas cells cultured on gelatin or plastic maintained the exclusively apical localization of MT1-MMP regardless of DDR1 signaling. This suggests that additional collagen-specific signaling besides that mediated through DDR1 contributes to regulating the apicobasal localization of MT1-MMP. A detailed understanding of how separate collagen-dependent signaling pathways converge to control the apicobasal localization of MT1-MMP warrants further investigation. It is unclear if the role of DDR1 in regulating MT1-MMP localization is directly related to its role in polarization. Interestingly, the ROCK target LIM kinase 1 (LIMK1) has been shown to control association of cortactin

with MT1-MMP–positive endosomes and, thus, increase the trafficking of MT1-MMP to invadopodia in epithelial breast cancer cells (Lagoutte et al, 2016). This raises the possibility that the ROCK-suppressing function of DDR1 could affect the localization of MT1-MMP through LIMK1.

Apicobasal polarity of epithelial cells in 2D culture was established and maintained under DDR1-IN-1 treatment where GP135 localized at the apical surface and intact junctional complexes formed in the expected locations. However, cytoskeletal organization and MT1-MMP localization were disturbed, potentially reflecting the phenotype observed upon inhibition of DDR1 in 3D culture. Interestingly, ectopic expression of dominant-negative mutant Rac1 (N17Rac1) which disrupts cell polarity in 3D does not affect cell polarity in 2D culture (O'Brien et al, 2001), and similar observations have been made for other gatekeepers of polarity in 3D including the Crumbs3-Pals1 complex (Roh et al, 2003), Par3 (Sfakianos et al, 2007; Horikoshi et al, 2009), and Cdc42 (Martin-Belmonte et al, 2007). Treatments with no apparent effect on epithelial polarity in 2D culture can, thus, disrupt polarity in 3D culture conditions, suggesting that the polarization process in 2D is more robust than in 3D culture.

We have extensively analyzed different epithelial organs in DDR1-null mice for potential defects in epithelial structures, including the lung, kidney, skin, and intestine. However, we did not find any notable defects (data not shown). The only difference was detected in the mammary gland epithelium during lactation where smaller alveoli and an altered pattern of basement membrane components were found in the DDR1-null mice. This may be explained by a temporal aspect to the role of DDR1 during establishment of polarity. To address whether DDR1 signaling is important for establishment and/or maintenance of polarity, we have examined the effect of DDR1-IN-1 at different time points during the formation of polarized structures. In contrast to the disruptive effects of the continuous presence of DDR1-IN-1 during cystogenesis (Fig 3), addition of DDR1-IN-1 after the establishment of polarized cysts in 3D culture did not influence the structures (data not shown), suggesting that DDR1 affects establishment rather than maintenance of polarity. Our TER measurements similarly indicated that DDR1 inhibition affects the initial establishment of polarity, but that the effect eventually is overcome. These data suggest that the lack of DDR1 in null mice may delay the polarization process, but that epithelial polarity eventually is established, which could explain the relatively mild phenotype of the DDR1-null mouse. The phenotype in the mammary gland of DDR1-null mice during puberty and pregnancy (Vogel et al, 2001) may, thus, be apparent because the mammary epithelium at these time points is under rapid reorganization which requires de novo polarization. This may also explain why we observe disorganized epithelial architecture and polarity-dependent basement membrane protein secretion in the lactating mammary gland, but not in other epithelial tissues. It is possible that triggering morphogenesis and matrix remodeling in other tissues (e.g., wound healing, induction of fibrosis) could result in other defects in the DDR1-null mouse.

DDR1 is highly expressed in different epithelial cell types, and the effect of DDR1 inhibition on polarity and morphogenesis in 3D matrix penetrates various epithelial cells, including MDCK cells, MCF10A cells, CaCO-2 cells, HC11 cells, and primary mouse mammary epithelial cells as we have shown in this study. Together with previous reports (Hidalgo-Carcedo et al, 2011; Rhys et al, 2018), this suggests that a fundamental role for DDR1, shared by various epithelial cells, may be to regulate actomyosin contractility at cell–cell junctions and thereby contribute to epithelial polarization and morphogenesis. Given that loss of polarity and aberrant tissue organization is associated with various pathologies including cancer, it will be valuable to gain an in depth understanding of this role of DDR1 signaling. It will be important to investigate how DDR1 is activated in the lateral cell membrane and how it regulates the phosphorylation status of MLC. In particular, the potential suppressive effect of DDR1 on the RhoA/ROCK pathway implicated by this study should be addressed in more detail to make molecular links in future.

## Materials and Methods

### cDNA cloning

FLAG (DYKDDDDK)-tagged MT1-MMP (MT1F) and mRFP-tagged MT1F have been described previously (Itoh et al, 1999; Weaver et al, 2014). The FLAG tag is inserted at the C terminus of the propeptide (between Arg111 and Tyr112) and is, thus, present at the N terminus of activated MT1-MMP. The mRFP-1 tag is inserted between Cys574 and Gln575 in the cytoplasmic domain with an addition of two-glycine N-terminal linker (Itoh et al, 2011). Construction of Ad-MT1F adenoviral vector using the AdEasyTM system (Q-BIOgene) has been described previously (Weaver et al, 2014). To generate MT1F-pHluorin-RFP, superecliptic pHluorin (Miesenbock et al, 1998) was inserted between Gly541 and Gly542 of MT1F-RFP with a two-glycine N-terminal linker and one-glycine C-terminal linker. MT1F-pHluorin-RFP was further subcloned into pSG5 (Stratagene) and pCEP4 (Invitrogen). The cDNA for superecliptic pHluorin was kindly provided by Prof Gero Miesenboeck (The Centre for Neural Circuits and Behaviour, University of Oxford).

### 2D and 3D culture of epithelial cells

We used four epithelial cell lines, including MDCK epithelial cells, MCF10A human mammary epithelial cells, CaCO-2 human colorectal carcinoma cells, and HC11 mouse mammary epithelial cells. MDCK lines used included clonal cells stably expressing DDR1 (MDCK-DDR1), dominant-negative DDR1 (MDCK-DDRΔC) or MT1F-pHluorin-RFP (MDCK-MT1pH cells). MDCK and CaCO-2 cells were cultured in DMEM containing 10% or 20% FBS and penicillin/streptomycin (P/S), respectively. MCF10A cells were cultured in DMEM/F12 supplemented with 5% FBS, P/S, 20 ng/ml EGF, 0.5 $\mu$g/ml hydrocortisone, 100 ng/ml cholera toxin, and 10 $\mu$g/ml insulin. HC11 cells were cultured in the RPMI 1640 medium supplemented with 10% FBS, 5 $\mu$g/ml insulin, 10 mM Hepes, and 10 ng/ml EGF. COS-7 cells were cultured in DMEM supplemented with 10% FBS and P/S.

For 2D culture of MDCK cells, the cells were seeded in culture inserts with 8-$\mu$m pores (24- or 12-well; BD Biosciences) or coverslips. For TER measurements, transwell inserts with 4-$\mu$m pores from Corning were used. Where indicated, coverslips or inserts were

coated with a thin layer of neutralized acid-extracted type I collagen (1.5 mg/ml; Nitta gelatin) and incubated at 37°C for 30 min to set. Alternatively, coverslips or inserts were coated with 50 μg/ml gelatin (unconjugated or labelled with Alexa dye). Unless otherwise indicated, the cells were cultured at 100% confluency.

In some experiments, cells were infected with Ad-MT1F at an MOI of five in the presence of GM6001 (10 μM; Elastin Products Company). 24 h after infection, GM6001 was washed out and the cells were stimulated with human HGF (50 ng/ml; Peprotech). To investigate the apicobasal distribution of MT1F, live cells were incubated with the FLAG M2 antibody (2 μg/ml) for 1 h in a chilled medium on ice to stain FLAG at the cell surface, 24 h after HGF stimulation. This was followed by fixation in 4% paraformaldehyde and further processing for immunofluorescence.

For 3D culture of all epithelial cell lines, the cells were suspended in neutralized acid-extracted collagen (2 mg/ml) and incubated for 1 h to allow the collagen to set. The cells were cultured in the presence or absence of HGF (50 ng/ml) and/or indicated inhibitors in a complete growth medium for up to 5 d. Inhibitors used include DMSO solvent control, GM6001 (10 μM), and DDR1-IN-1 (1 μM). DDR1-IN-1 (Kim et al, 2013) was kindly provided by Dr. Nathanael Gray (Dana Farber Cancer Institute, Harvard Medical School, USA). 3D culture on microcarrier beads was adapted based on a fibroblast invasion assay described by Palmisano and Itoh (Palmisano & Itoh, 2010). Gelatin-coated Cytodex3 microcarrier beads (Sigma-Aldrich) were incubated with cells suspended in a complete growth medium at 37°C with gentle agitation on the shaker (45 rpm). After overnight incubation, it was confirmed by microscopy that the cells had formed confluent layers on the beads. Cell-coated beads were then rinsed and gently resuspended in neutralized, acid-extracted collagen (2 mg/ml) containing growth factors/inhibitors as indicated. The gels were allowed to set for 1 h at 37°C before overlaying gels with a complete growth medium and cultured for 3–5 d. For MCF10A cells, EGF was excluded from the culture medium. The medium was replaced every second day.

## Branching morphogenesis assay with primary mammary organoids

The branching morphogenesis assay with primary mammary organoids was adapted from a previous protocol (Mroue & Bissell, 2013). Two female C57Bl/6J mice between the age of 12 and 14 wk were euthanized in a gradient of $CO_2$. The second, third, and fourth pair of mammary glands were dissected out and rinsed in ice-cold PBS. The lymph nodes in the fourth pairs of mammary glands were removed, before mincing gland tissue with a scalpel. Glands were transferred to a collagenase solution (DMEM/F12, 5% FBS, and insulin [5 μg/ml], trypsin [2 mg/ml], and collagenase [2 mg/ml]) and incubated 1 h at 37°C with agitation. Epithelial organoids were then pelleted by centrifugation at 520 $g$ for 10 min at 25°C. The fatty layer on top and the aqueous phase was removed and pellets resuspended in DNase solution (DMEM/F12 with DNase I [4 U/ml]). Epithelial fragments were then pelleted by centrifugation (520 $g$). Differential centrifugations were performed to wash out enzymes and unwanted single cells. Purified organoids were resuspended in 1:2 Matrigel/collagen I (neutralized, acid-extracted collagen) at a concentration of 1,000 organoids/ml. The gels were set at 37°C for

30 min before overlaying with the organoid medium (DMEM/F12 medium with 1× ITS supplement (insulin, transferrin, and selenium). Basic FGF (2.5 nM), DDR1-IN-1 (0.5 μM), and GM6001 (10 μM) were included as indicated. The medium was renewed every second day. Progress of tubular morphogenesis was monitored by widefield imaging throughout the well between day 5 and 7.

## Lactogenic differentiation of HC11 cells to form mammospheres

Mammosphere formation assays where performed as previously described (Morrison & Cutler, 2009). Briefly, HC11 cells were seeded at 100% confluence in six-well plates and cultured for 5 d in a complete growth medium with appropriate inhibitors. The next day, the medium was replaced with the EGF-free medium. 24 h later, the medium without EGF but containing dexamethasone (10 μM), insulin (5 μg/ml), and prolactin (5 μg/ml) (DIP medium) was added to induce lactogenic differentiation. Mammosphere formation was monitored by widefield microscopy throughout the wells on day 3, 4, and 5 after addition of the DIP medium. After 5 d, cells were lysed for extraction of total RNA or protein.

## Indirect immunofluorescence staining

Cells in 2D culture were fixed in 4% PFA and blocked with 5% goat serum and 3% BSA in TBS (blocking solution) before incubation with primary antibodies in blocking solution. After thorough washing in PBS, the samples were incubated with Alexa-conjugated secondary antibodies produced in goat (Molecular Probes; Thermo Fisher Scientific), Alexa-conjugated phalloidin (Molecular Probes; Thermo Fisher Scientific), and DAPI. For staining surface MT1F, live cells were incubated with the FLAG M2 antibody on ice for 30 min as described above. A blocking step was then performed before sequential incubation with Alexa-conjugated secondary antibodies and Alexa-conjugated phalloidin in the presence of 0.1% Triton. After extensive washes in PBS, coverslips and membranes cut from culture inserts were and mounted on glass slides using the Prolong Gold/Diamond Anti-fade mounting medium (Molecular Probes; Thermo Fisher Scientific).

For staining of cells in 3D collagen gels, cells (either as single suspended cells or cultured on beads) were cultured in 100 μl of 3D collagen gel (2 mg/ml) as described above. The whole gel was fixed in 4% PFA in PBS for 30 min at room temperature on a shaker. The wells and inserts were rinsed three times with PBS before 1-hr incubation in a permeabilization/blocking solution (0.5% Triton X-100 and 10% FBS in PBS) on a shaker at room temperature. Primary antibodies diluted in permeabilization solution were added to the wells and inserts, and the plates were incubated on a shaker overnight at 4°C. After thorough washes in permeabilization solution at room temperature, the gels were incubated for 4 h with secondary Alexa-conjugated antibodies (Molecular Probes; Thermo Fisher Scientific), Alexa-conjugated phalloidin (Molecular Probes; Thermo Fisher Scientific), and DAPI in a permeabilization solution. After thorough washes in the permeabilization solution, a final wash in PBS was performed before cutting the membranes from inserts and mounting gel on glass slides using the Prolong Gold/Diamond Anti-fade mounting medium (Molecular Probes; Thermo Fisher Scientific).

The following primary antibodies were used for staining: acetylated tubulin (6-11B-1; Sigma-Aldrich), DDR1 (sc-532; Santa Cruz), E-cadherin (BD Biosciences), FLAG M2 (F1804; Sigma-Aldrich), GM130 (ab52649; Abcam), GP135 (3F2/D8 hybridoma supernatant), laminin (L9393; Sigma-Aldrich), ppMLC (Thr18/Ser19, 36745; Cell Signaling), and ZO-1 (33-9100, ZYMED; Thermo Fisher Scientific).

### Immunohistochemistry (IHC)

Wild-type and DDR1-null C57Bl/6J female mice of 13–14 wk age were mated with wild-type males. 3 d after giving birth, the female was euthanised in a rising gradient of $CO_2$. The fourth pair of mammary glands was dissected out and rinsed in ice-cold PBS before fixation in formalin. The tissue was embedded in paraffin and sections cut. Sections were deparaffinised and rehydrated before quenching endogenous peroxidase activity by 20-min incubation in $H_2O_2$ in $dH_2O$. Antigen retrieval was performed by heating slides in citrate buffer (0.01 M trisodium citrate dehydrate and 0.05% Tween, pH 6) in a 92°C water bath. The slides were incubated 2 h in blocking buffer (10% normal goat serum in TBS) before overnight incubation with primary antibodies at 4°C. The next day, the sections were washed in PBS and incubated with biotinylated secondary antibodies (Vector Laboratories) in a blocking solution at room temperature for 2 h. After extensive washes in PBS, the slides were incubated 30 min with avidin/biotin/HRP complex reagent (ABC staining kit; Vector Laboratories) before washing and developing with DAB substrate (Vector Laboratories). The sections were counterstained with hematoxylin, dehydrated, and mounted using the DPX mounting medium (Sigma-Aldrich). The following primary antibodies were used: collagen IV (ab6586; Abcam) and laminin (L9393; Sigma-Aldrich).

### Image acquisition

All widefield images were captured on an inverted Nikon TE2000-E widefield microscope with Volocity Acquisition software (PerkinElmer). Following objective lenses were used: 4× objective lens (Plan Fluor 4×/NA 0.13), 10× objective lens (UPLSAPO 10×/NA 0.30 DIC), and 20× objective lens (UPLSAPO 20×/NA 0.45). Confocal laser scanning microscope imaging used a Fluoview Spectral FV1200 CSLM based on Olympus IX83 motorized inverted microscope CSLM. Following objective lenses were used: 60× objective lens (UPLSAPO 60×/NA 1.4) and a 30× objective (UPLSAPO 30×/NA 1.05 Silicone Immersion). Spinning disc confocal microscopy imaging was performed on a PerkinElmer Spinning Disk Confocal Microscope based on a Nikon TE 2000-U Eclipse motorized inverted microscope with DIC optics. The following objective lens was used: 60× (Plan Apo 60×/NA 1.40). Volocity software (PerkinElmer) was used for Acquisition.

### Image data analyses

Volocity (PerkinElmer), Adobe Photoshop (Adobe, Berkshire), and ImageJ were used for image processing, visualization, and analysis. The same adjustments were applied to all images within the same experiment. Original intensity values were used for quantification.

To calculate the proportion of MT1F that localized to the basal surface, fluorescence intensities (FI) of apical and basal surfaces in 10–20 confocal stacks per condition were analyzed by Volocity measurement module software, and the proportion of basal localization (%) was calculated as follows: 100*basal-FI/(basal-FI + apical-FI). To find the proportion of MT1F-pHluorin-RFP at the basal surface, ImageJ was used to measure FI in the 488-nm channel at the apical and basal surface in more than 30 confocal images, and data analysis was otherwise performed as above. The mean proportion of basal surface MT1-MMP were compared between the different conditions with a Kruskal test before localizing differences with a pairwise Wilcoxon test with Holm correction of the *P*-value to correct for multiple testing.

To measure the protrusion length and area in the tubulogenesis assay on microcarrier beads, images were captured by bright-field imaging after 3 d culture in 3D collagen I matrices. Protrusion length was quantified by measuring the length of the 10 longest protrusions on five structures per condition with the starting point defined as the point in the outer layer of cells from which the tubular structure extends. Measurements were pooled and comparison between conditions done with a Kruskal test (nonparametric alternative to ANOVA) before localizing differences with a pairwise Wilcoxon test with Holm correction of the *P*-value to correct for multiple testing. The area of the structures was measured using ImageJ software, using the outermost periphery of the structures as the border. Similarly, the area of the beads was measured using ImageJ software. The area of the structures was normalized to the area of bead to compensate for variations in bead size and thereby variation in the starting number of cells and area. 5–7 structures per condition were used for quantification, and the ratios of structure/bead area were compared between the different conditions with a Kruskal test before localizing differences with a pairwise Wilcoxon test with Holm correction of the *P*-value to correct for multiple testing.

To evaluate mammosphere formation, bright-field images were captured by widefield microscopy on day 4 after addition of the DIP medium. 10–20 fields were captured in each well and the number of mammospheres per field counted to compare mammosphere formation between different conditions. The average mean number of mammospheres per field from three independent experiments was compared between the tested conditions with a Kruskal test before localizing differences with a pairwise Wilcoxon test with Holm correction of the *P*-value to correct for multiple testing.

Branching morphogenesis by primary, mammary organoids was evaluated by capturing bright-field images throughout the well and categorising organoids in 33–70 images per condition as either undergoing branching morphogenesis or not based on the appearance of three or more branches. The percentage of organoids undergoing tubular morphogenesis was calculated for each condition, and the mean of three independent experiments was compared.

Tissue sections processed for IHC were analyzed by widefield microscopy and white balance adjusted by applying the same filter to all images in Photoshop (Adobe). ImageJ was used for automatic detection and quantification of alveolar area in 5–10 sections per mouse. The mean alveolar size was found for three mice per genotype and compared with a Kruskal test before localizing

differences with a pairwise Wilcoxon test with Holm correction of the *P*-value to correct for multiple testing.

R software (The R Foundation) was used for data visualization and statistical analysis.

## Collagen film degradation assay

Collagen film degradation assay was performed as described previously (Itoh et al, 2006), with slight modifications. MDCK cells (1 × 10$^6$ cells/well) were cultured in a 12-well CellBind (CoStar) plate coated with collagen I (1:1 ratio of protease-extracted and acid-extracted collagen I). The cells were then infected with Ad-MT1F adenovirus at an MOI of five in the presence of GM6001 (10 $\mu$M). After 24 h culture, GM6001 was washed out and the cells incubated for a further 16 h in the presence or absence of HGF (50 ng/ml) and relevant inhibitors. After 24 h of culture, the cells were removed by trypsinization, and the remaining collagen was fixed with 4% PFA and stained with Coomassie blue. Bright-field images were captured to visualize degraded collagen as white areas against grey collagen. Photoshop (Adobe) was used to adjust white balance and contrast. The same filter was applied to all images to allow comparison.

## TER measurements

MDCK cells were seeded at 100% confluence in culture inserts with 0.4-$\mu$m pores (Corning; Sigma-Aldrich) in the presence of relevant inhibitors. 24 h after seeding, TER was measured with a sterile electrode using and epithelial volt ohm meter (World Precision Instruments (Matter & Balda, 2003). Measurements were repeated three times per day. MDCK cells were trypsinized and resuspended in a low-Ca$^{2+}$ medium (Sigma spinner culture medium with 2 mM glutamine, 1 mM sodium pyruvate, 10% dialyzed FBS, and P/S) to suppress E-cadherin–mediated cell–cell adhesion. The cells were then seeded in culture inserts in a low-Ca$^{2+}$ medium. The next morning, the medium was replaced with a complete growth medium and TER measurements initiated after 15 min incubation. Measurements were repeated at 1-hr intervals. Three measurements were done for all wells at each time point and the mean value noted. TER of a collagen-coated/uncoated insert without cells was used as the background value. Experiments were performed in triplicate and the mean TER value and SD of three wells plotted.

## Immunoprecipitation of DDR1

DDR1 was immunoprecipitated as described previously (Shitomi et al, 2015) with slight modifications. MDCK cells were cultured in a serum-free medium overnight before 4-h stimulation with 100 $\mu$g/ml protease-extracted collagen I. The cells were lysed in the RIPA (radio imunoprecipitation assay) buffer (1% Triton X-100, 0.1% SDS, 1% deoxycholic acid, 0.02% NaN$_3$, 50 mM Tris–HCl, pH 8.0, 150 mM NaCl) with freshly added sodium orthovanadate, protease inhibitor cocktail, and Marimastat to prevent shedding of DDR1. The samples were then sonicated and centrifuged to precipitate debris. Supernatants were pre-cleared by incubation with protein G magnetic Dynabeads (Thermo Fisher Scientific) on a rotor for 1 h at 4°C to remove nonspecifically binding proteins from the lysates. The samples were then incubated with 5 $\mu$g of DDR1 antibody (sc-532; Santa Cruz) overnight at 4°C on a rotor. Next day, Dynabeads were added to capture DDR1 antibody and the sample incubated for 2 h at 4°C on a rotor. The beads were washed with RIPA buffer and TBS with protease inhibitor and sodium orthovanadate before resuspending the beads in 1× SDS buffer. Total lysate and eluate were analyzed by Western blot.

## SDS–PAGE and Western blotting

Total cell lysates were prepared by the addition of 1× SDS–PAGE loading buffer containing 2-mercaptethanol to the cells in the culture plate and subsequent boiling for 20 min. The cell lysates and immunoprecipitated materials were separated by SDS–PAGE, and the proteins in the gel were transferred onto a polyvinylidene fluoride membrane using a Trans-Blot Turbo Transfer System (Bio-Rad). After blocking of the membrane with 10% skim milk in TBS, the membrane was probed with primary antibodies. Proteins were visualized using alkaline phosphatase–conjugated secondary antibodies (DAKO) and Western Blue alkaline phosphatase substrate (Promega). The membrane was dried and scanned using a Canon CanoScan LiDE 210 scanner at 600 DPI. Primary antibodies used were anti-actin (sc-1616; Santa Cruz), anti-DDR1 (cytoplasmic domain, sc-532; Santa Cruz), anti-DDR1 (ectodomain, AF2396; RD Systems), anti-MLC (ab92721; Abcam), anti-MT1-MMP (loop region, EP1264Y; Abcam), anti-ppMLC (Thr18/Ser19, 36745; Cell Signaling), and anti-phosphor-tyrosine (4G10; Millipore).

## RT-PCR to evaluate mRNA levels of *β*-casein in HC11 cells after lactogenic differentiation

HC11 cells were subjected to lactogenic differentiation as described above. After 5 d in the DIP medium, total RNA was extracted using the RNAqueous Micro kit (Invitrogen; Thermo Fisher Scientific) according to the manufacturer's instructions. RNA was reversed-transcribed using the High Capacity cDNA Reverse Transcription kit from AB applied Biosciences (Thermo Fisher Scientific). cDNA was then used as a template for PCR using the Dream Taq Polymerase (Thermo Fisher Scientific). Primers for GADPH were used as a housekeeping gene for normalization to the total amount of RNA. The following primers were used:

Primers: *β*-casein:
 Fwd: GACTACATTTACTGTATCCTCTGAG
 Rev: GTGCTACTTGCTGCAGAAAGTACAG

Primers GADPH (canine):
 Fwd: TTCACCACCATGGAGAAGGC
 Rev: GGTCCCTCCGATGCCTGC.

## Statistics and reproducibility

R software (The R Foundation) was used for statistical analysis and data visualization. For all parametric data, a *t* test was used. Nonparametric data were analyzed using the Wilcoxon's rank sum test. In case of multiple comparisons, ANOVA (parametric

data) or Kruskal–Wallis (nonparametric data) tests were used followed by post hoc pairwise tests with Bonferroni–Holm's correction for multiple testing. Unless otherwise stated in the figure legends, results are representative of at least three independent experiments.

## Supplementary Information

## Acknowledgements

We thank Dr. Nathanael S Gray for providing DDR1-IN-1 and Prof. Gero Miesenboeck for providing superecliptic pHluorin cDNA. This study was supported by Kennedy Trust of Rheumatology Research Dphil studentship (PP Søgaard and Y Itoh) and BBSRC research grant BB/N014855/1 (K Matter).

### Author Contributions

PP Søgaard: conceptualization, data curation, formal analysis, validation, investigation, visualization, methodology, and writing—original draft, writing—review, and editing.
N Ito: data curation, formal analysis, and investigation.
N Sato: data curation, validation, and investigation.
Y Fujita: supervision and writing—review and editing.
K Matter: conceptualization, supervision, funding acquisition, methodology, and writing—original draft, review, and editing.
Y Itoh: conceptualization, resources, formal analysis, supervision, funding acquisition, investigation, visualization, methodology, project administration, and writing—original draft, review, and editing.

### Conflict of Interest Statement

The authors declare that they have no conflict of interest.

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
