## [Reviewer comments · Life Science Alliance]

Life Science Alliance

Epithelial polarization in 3D matrix requires DDR1 signaling to regulate actomyosin contractility

Pia Sogaard, Noriko Ito, Nanami Sato, Yasuyuki Fujita, Karl Matter, and Yoshifumi Itoh
DOI: <https://doi.org/10.26508/lsa.201800276>

Corresponding author(s): Yoshifumi Itoh, University of Oxford

Review Timeline:	Submission Date:	2018-12-11
	Editorial Decision:	2018-12-11
	Revision Received:	2019-02-06
	Editorial Decision:	2019-02-07
	Revision Received:	2019-02-07
	Accepted:	2019-02-08

Scientific Editor: Andrea Leibfried

Transaction Report:

Please note that the manuscript was previously reviewed at another journal and the reports were taken into account in the decision-making process at Life Science Alliance.

Reviewer #1 Review

Comments to the Authors (Required):

This manuscript begins by documenting a collagen dependent re-localization of MMP14 (MT1-MMP) in response to HGF. The authors identify that DDR1 might be the relevant collagen receptor and then perform a series of experiments demonstrating that DDR1 is required for collective invasion in 3D models containing collagen. Of note, both the kinase activity and C-terminus of DDR1 are required for its control of invasion. The authors then proceed to show that inhibition of DDR1 kinase activity leads to altered actomyosin contractility and a gain in ROCK-dependent pMLC proximal to cell-cell junctions. Inhibition of ROCK restores invasion when DDR1 is inhibited. The authors present analysis of DDR1 knockout mammary glands and a mammary differentiation model. This provides functional data to support their in vitro observations of DDR1 antagonizing pMLC. Overall, there are many interesting facets to this work and the data are generally convincing. With some additional experimental work and tidying up of the narrative, this manuscript could be suitable for the Journal of Cell Biology.

Specific comments

1. The paper is a bit disjointed, it has a strong focus on MMP14 to begin with that then disappears and the reverse situation with the analysis of pMLC. The in vivo and differentiation experiments should also include analysis of MMP14 and it would be worth knowing if the ROCK/MLC axis is required for the changes in MMP14 localization. To address the first issue, the authors should stain the mouse tissue for MMP14 and with a cleaved collagen antibody that recognizes the product of MMP14 activity.
2. There is a potential paradox in the role of DDR1 that is not fully explored. The fact that inhibition of DDR1 phenocopies HGF treatment with regards MMP14 localization suggests that HGF might inhibit DDR1 function. In contrast, the fact that DDR1 is required for HGF induced invasion suggests that HGF might positively influence DDR1. The authors should test whether HGF alters DDR1 phosphorylation on collagen I or if it re-localizes DDR1. Also, more discussion of this matter would be welcome.
3. The ΔC term DDR1 experiments are interesting, but are not carried through into the second part of the study involving pMLC. This should be rectified.
4. What happens if the authors express a kinase dead DDR1, as opposed to using the inhibitor?
5. The manuscript would be greatly enhanced by some more details about the proximal binding partners of DDR1 that control polarity and pMLC. Links to Par3 and Par6 have been previously reported (Chow et al., and Hidalgo-Carcedo et al.), are they relevant in this system? The authors mention a lack of co-IP between DDR1 and Par3, but they should test whether Par3 and/or Par6 localization is disrupted by either the DDR1 inhibitor or the ΔC terminal construct. The latter point is pertinent because the C terminus of DDR1 is reported to bind PDZ domains. Further, what are the DDR1 substrates that are required for polarity regulation? Or if it is only auto-phosphorylation, then what are the relevant phospho-tyrosine binding proteins that are recruited? There is some speculation in the discussion, but no clear evidence.
6. Figure 9d is not convincing.
7. The manuscript is a long read and should be shortened. Figures 3&4 could be merged and Figure 5&6 could probably be condensed.

Reviewer #2 Review

Comments to the Authors (Required):

The manuscript by Sogaard et al examines the role of the collagen receptor kinase DDR1 in polarisation and tubulogenesis of cell lines and organoids in 3-Dimensional culture systems. The authors report that DDR1 plays a role in controlling actomyosin contractility and cell-cell junctions. The authors posit that this controls the orientation of cell polarity required for morphogenetic rearrangements, such as HGF-induced tubulogenesis of MDCK cysts.

Although the majority of this work is clear, it does not provide a clear extension of our understanding of DDR1 function in polarisation. The work presents as a series of (clear and well-written) phenomenological observations, in myriad and occasionally disjointed, model systems. It is not clear how phenotypes are linked between these systems.

A function for DDR1 in HGF-induced tubulogenesis has already been reported (Wang et al, J Cell Physiol, 2005.). Moreover, that inhibition of Rho-ROCK signalling pathways can correct DDR1 deficiencies, and polarity defects, are already well-known. There, the work does not provide an advance into understanding the mechanisms of polarisation, DDR1 function, or MT1-MMP function. As the authors themselves note, whole body knockout of DDR1 does not show large-scale polarisation defects. It is hard, therefore, to rationalise that (arguably) minor increments of understanding warrant publication in this journal.

The authors are commended on a well-written and clear article. It is unfortunate that it does not provide sufficient advance to warrant the journal level.

Reviewer #3 Review

Comments to the Authors (Required):

Overall evaluation:

The finding described here can be summarized as: (1) inhibition of the collagen-activated RTK DDR1 interferes with lumen formation in 3D but not 2D MDCK cell cultures and does so also in a mammary epithelia-derived cell line and in mammary organoids. (2) pharmacological RhoK/MyoII inhibition can overcome this defect. The authors complement these findings by showing that previously characterized defects in mammary duct development in DDR1-KO mice are associated with disrupted collagenIV/laminin deposition similar to their observation in MDCK cells.

No mechanism is forwarded, however. DDR1 is known, on the one hand, to co-localize and to interact with E-cadherin and on the other hand, to mediate Myosin II-dependent contraction at cell-ECM adhesion sites; how DDR1 gets activated at the MDCK cell-cell junctions, how it signals there for RhoK/MyoII activation and how exactly RhoK/MyoII activity at cell-cell junctions regulates 3D morphogenesis was not investigated. That RhoA activity contributes to lumen formation in MDCK cells has been reported several times before and the mammary gland defect in DDR1 KO mice has also been previously described. In my opinion, at this state of knowledge a more mechanistic advance could be expected for a paper in this journal.

Specific suggestions/comments:

I believe data presentation could be improved:

The authors start out analyzing HGF-mediated tubulogenesis, the most complex of the MDCK polarization assays, when the lumen defect is apparent in the cysts that form in simple 3D cultures. They also don't properly introduce their tubulogenesis assay and the distinct signaling steps it is known to involve, talking instead about "stimulation with morphogen".

Figs 2 and 3 are largely redundant, with Fig. 3 being far more informative. The MT1-MMP data do not connect to the rest of the findings. It remains unclear if the apical-to-basal MMP redistribution in 2D collagen cultures bears any significance for the polarity phenotype in the cysts.

In my opinion, the Figure legends are too detailed in the methodology.

Additional comments:

The references Meder et al. and Torkko et al., are not the original reports that introduced Gp135 as MDCK cell apical marker and the cilium as a differentiation marker, respectively.

Fig.S1: what are the multiple bands and arrowheads pointing to them?

Fig. 5A: the E-cadherin staining is poor

Fig.6B needs some ppMLC quantitation because there are stronger and weaker labeled junctions apparent in both conditions. In the x-z view in Fig.6C it appears that ctrl and DDR1-inhibited cells differ primarily in apical rather than junctional pMLC. The apical surface of polarized MDCK cells is not usually under myosin tension. Please explain what the apical contractile structures are. How was the specificity of the ppMLC staining controlled? In Fig.6B a lot of the labeling appears to be diffusely intracellular and not obviously associated with fibers.

December 11, 2018

Re: Life Science Alliance manuscript #LSA-2018-00276-T

Dr. Yoshifumi Itoh
University of Oxford
Kennedy Institute of Rheumatology
65 Aspenlea Road
Hammersmith
London, Oxon W6 8LH
United Kingdom

Dear Dr. Itoh,

Thank you for transferring your manuscript entitled "Epithelial polarization in 3D matrix requires DDR1 signaling to regulate actomyosin contractility" to Life Science Alliance. The manuscript was assessed by expert reviewers at another journal before, and the editors have transferred those reports to us with your permission.

The reviewers who assessed your manuscript appreciated the work, but would have expected further reaching mechanistic insight into how DDR1 affects cell polarization and tubulogenesis. This is not a concern for publication in Life Science Alliance, and I would thus like to invite you to submit a revised version to address the specific comments made by the reviewers. We would expect a point-by-point response and accordingly text changes / representation changes including the requested quantifications. No additional experiments are needed.

The typical timeframe for revisions is three months.

Thank you for this interesting contribution to Life Science Alliance. We are looking forward to receiving your revised manuscript.

Sincerely,

Andrea Leibfried, PhD
Executive Editor
Life Science Alliance

Meyerhofstr. 1
69117 Heidelberg, Germany
t +49 6221 8891 502
e a.leibfried@life-science-alliance.org
www.life-science-alliance.org

- A letter addressing the reviewers' comments point by point.
- An editable version of the final text (.DOC or .DOCX) is needed for copyediting (no PDFs).
- High-resolution figure, supplementary figure and video files uploaded as individual files: See our detailed guidelines for preparing your production-ready images, <http://life-science-alliance.org/authorguide>
- Summary blurb (enter in submission system): A short text summarizing in a single sentence the study (max. 200 characters including spaces). This text is used in conjunction with the titles of papers, hence should be informative and complementary to the title and running title. It should describe the context and significance of the findings for a general readership; it should be written in the present tense and refer to the work in the third person. Author names should not be mentioned.

B. MANUSCRIPT ORGANIZATION AND FORMATTING:

Full guidelines are available on our Instructions for Authors page, <http://life-science-alliance.org/authorguide>

Responses to the reviewer's comments

Reviewer #1

This manuscript begins by documenting a collagen dependent re-localization of MMP14 (MT1-MMP) in response to HGF. The authors identify that DDR1 might be the relevant collagen receptor and then perform a series of experiments demonstrating that DDR1 is required for collective invasion in 3D models containing collagen. Of note, both the kinase activity and C-terminus of DDR1 are required for its control of invasion. The authors then proceed to show that inhibition of DDR1 kinase activity leads to altered actomyosin contractility and a gain in ROCK-dependent pMLC proximal to cell-cell junctions. Inhibition of ROCK restores invasion when DDR1 is inhibited. The authors present analysis of DDR1 knockout mammary glands and a mammary differentiation model. This provides functional data to support their in vitro observations of DDR1 antagonizing pMLC. Overall, there are many interesting facets to this work and the data are generally convincing. With some additional experimental work and tidying up of the narrative, this manuscript could be suitable for the Journal of Cell Biology.

Specific comments

1. The paper is a bit disjointed, it has a strong focus on MMP14 to begin with that then disappears and the reverse situation with the analysis of pMLC. The in vivo and differentiation experiments should also include analysis of MMP14 and it would be worth knowing if the ROCK/MLC axis is required for the changes in MMP14 localization. To address the first issue, the authors should stain the mouse tissue for MMP14 and with a cleaved collagen antibody that recognizes the product of MMP14 activity.

As the reviewer pointed out, the paper was slightly disjointed. However, we decided to do it this way as this is the process that has led us to discover the role of DDR1 in epithelial polarity. However, to facilitate reading, we have modified the text. Regarding the analysis of MMP14 in mouse tissue and differentiation models, MMP14 is not functional in mouse mammary epithelial cells and the differentiation model. Thus, we believe that it is the best to concentrate on polarity aspects excluding MMP14 in the mouse model.

2. There is a potential paradox in the role of DDR1 that is not fully explored. The fact that inhibition of DDR1 phenocopies HGF treatment with regards MMP14 localization suggests that HGF might inhibit DDR1 function. In contrast, the fact that DDR1 is required for HGF induced invasion suggests that HGF might positively influence DDR1. The authors should test whether HGF alters DDR1 phosphorylation on collagen I or if it re-localizes DDR1. Also, more discussion of this matter would be welcome.

This is indeed a complex part of our manuscript, but we do not think there is a paradox. It does look like that DDR1-IN-1 treatment phenocopies HGF treatment when basal localization of MMP14 is monitored. However, there is a major difference: DDR1-IN-1 does not enhance epithelial cell invasion. Together with our data on transepithelial resistance, we think that DDR1 kinase activity is essential for sorting molecules to the different membrane domains, which is crucial for proper polarization process. Although DDR1-IN-1 treatment does not completely disrupt polarised localization of GP135, E-cadherin and ZO1 in 2D conditions, the data indicate that this apparently polarized epithelium is not functional. We have also confirmed that HGF does not influence DDR1 phosphorylation. We now added this notion in the result section explaining Figure 2.

3. The ΔC term DDR1 experiments are interesting, but are not carried through into the second part of the study involving pMLC. This should be rectified.

We showed that DDR1 knockdown, overexpression of DDR1 ΔC and pharmacological inhibition of DDR1 by DDR1-IN-1 resulted in exactly the same phenotypes. Although the data is not shown, based on morphology of the cells we expect that DDR1 ΔC -expressing cells show the same phenotype as DDR1-IN-1 treated cells in terms of ppMLC. Since DDR1 ΔC expression phenocopies DDR1-IN-1 treatment, and it may be criticized as overexpression of dominant negative mutant (DDR1 ΔC may exhibit potential non-specific side effects),

we did not add these data. We believe that showing DDR1-IN-1 data supports the conclusion we made well.

4. What happens if the authors express a kinase dead DDR1, as opposed to using the inhibitor?

DDR1 Δ C is a dominant negative mutant which inhibits collagen-induced DDR1 phosphorylation (Wang et al J.Cell Physiol, 2005). Thus, expression of DDR1 Δ C mimics DDR1 knockdown and DDR1-IN-1 treatment in our study. On the other hand, kinase-dead DDR1 (DDR1KD) does not act as a dominant negative mutant, since despite lack of its kinase activity DDR1KD can be phosphorylated to transmit the signal by forming a complex with endogenous DDR1 upon collagen stimulation (Juskaite et al. Elife, 2017). We have preliminary data showing that overexpression of DDR1KD does not have impact on tubulogenesis by MDCK cells in 3D collagen. However, this notion is not relevant to the context of this manuscript, and thus we do not discuss this.

5. The manuscript would be greatly enhanced by some more details about the proximal binding partners of DDR1 that control polarity and pMLC. Links to Par3 and Par6 have been previously reported (Chow et al., and Hidalgo-Carcedo et al.), are they relevant in this system? The authors mention a lack of co-IP between DDR1 and Par3, but they should test whether Par3 and/or Par6 localization is disrupted by either the DDR1 inhibitor or the Δ C terminal construct. The latter point is pertinent because the C terminus of DDR1 is reported to bind PDZ domains. Further, what are the DDR1 substrates that are required for polarity regulation? Or if it is only auto-phosphorylation, then what are the relevant phospho-tyrosine binding proteins that are recruited? There is some speculation in the discussion, but no clear evidence.

As the reviewer mentioned, we speculate that Src homology-2 domain-containing phosphotyrosyl phosphatase 2 (SHP-2) is a potential downstream signal mediator of DDR1 for cell polarity. It has been previously shown that SHP-2 is activated downstream of DDR1 (Wang et al., 2006) and has been shown to negatively regulate RhoA through inhibition of Vav2, a Rho guanine nucleotide exchange factor (GEF) in MDCK cells (Kodama et al., 2000). It was also shown that SHP-2 interacts with phosphorylated DDR1 (Wang et al., 2006) and a DDR1 phosphopeptide pull-down assay has suggested Vav2 to interact with pY484 on DDR1 (Lemeer et al., 2012) raising the possibility that DDR1 by acting as a scaffold for both proteins could locally enhance SHP-2-mediated inhibition of Vav2. It is thus possible that SHP-2 activated downstream of DDR1 signaling at the lateral membrane suppresses RhoA/ROCK signaling through inhibition of Vav2 and inhibition of DDR1 would therefore result in an increase in Rho/ROCK activity along cell-cell contacts. We added this notion in the discussion section.

6. Figure 9d is not convincing.

We repeated this experiment five times, and relative ppMLC band intensity over MLC band were measured. We added statistical analyses of five experiments as a graph. The data indicate that DDR1-IN-1 treatment significantly increased ppMLC ($p < 0.03$) and further treatment with H115 significantly decreased ppMLC levels ($p < 0.015$). We hope this analysis make the data convincing.

7. The manuscript is a long read and should be shortened. Figures 3&4 could be merged and Figure 5&6 could probably be condensed.

We agree with the reviewer. We have now shortened the text and merged Figures 2 and 3 to make a new Figure 2, and Figures 5 and 6 to make a new Figure 4.

Reviewer #2

The manuscript by Sogaard et al examines the role of the collagen receptor kinase DDR1 in polarisation and tubulogenesis of cell lines and organoids in 3-Dimensional culture systems. The authors report that DDR1 plays a role in controlling actomyosin contractility and cell-cell junctions. The authors posit that this controls the orientation of cell polarity required for morphogenetic rearrangements, such as HGF-induced tubulogenesis of MDCK cysts.

Although the majority of this work is clear, it does not provide a clear extension of our understanding of DDR1 function in polarisation. The work presents as a series of (clear and well-written) phenomenological observations, in myriad and occasionally disjointed, model systems. It is not clear how phenotypes are linked between these systems.

We use a variety of epithelial 3D culture systems in our study including MDCK, CaCO-2 and MCF10A cells as well as primary mammary organoids. Additionally HC11 cells are used in the semi-3D mammosphere formation assay. Regardless of different model systems the establishment of epithelial polarity is a fundamental requirement, and our data indicate that DDR1 kinase activity is essential in this process. The data from transepithelial resistance assay indicate that DDR1 plays a major role in sorting the molecules to the different membrane domains, which is a fundamental and links all of these model systems.

A function for DDR1 in HGF-induced tubulogenesis has already been reported (Wang et al, J Cell Physiol, 2005.). Moreover, that inhibition of Rho-ROCK signalling pathways can correct DDR1 deficiencies, and polarity defects, are already well-known. There, the work does not provide an advance into understanding the mechanisms of polarisation, DDR1 function, or MT1-MMP function. As the authors themselves note, whole body knockout of DDR1 does not show large-scale polarisation defects. It is hard, therefore, to rationalise that (arguably) minor increments of understanding warrant publication in this journal.

It is unfortunate that the reviewer felt that our manuscript did not have enough increment of knowledge for this journal. Here I wish to reemphasize our discoveries.

1. Regulation of the apicobasal localization of MT1-MMP requires both DDR1-dependent and independent collagen signaling. (Novel)
2. We discovered that DDR1, which is expressed in all epithelial cells, plays a crucial role during the establishment of epithelial polarity in MDCK cells. This was corroborated in MCF10A, CACO-2 and mouse mammary epithelial cells both *in vitro* and *in vivo*. This suggests that DDR1-mediated polarization is a general fundamental mechanism in epithelial cells. (Novel)
3. We have shown that DDR1 kinase inhibition inhibited establishment of epithelial polarity in 3D collagen matrices and that this phenotype can be recovered by ROCK inhibitors, indicating that DDR1 signalling inhibits Rho-ROCK activity in a spatial manner. (Novel) Although the reviewer mentioned that this is "well-known", this is the first report demonstrating that the kinase activity of DDR1 guides Rho/ROCK signalling during cell polarization.

It is true that our discoveries somehow overlap with reports by Sahai's group (Hidalgo-Carcedo et al, Nat Cell Biol, 2011) and Godinho's group (Rhys et al, J Cell Biol, 2018), where they suggested a link between DDR1 and Rho-ROCK dependent actomyosin contraction. However, their observations regard biological processes distinct from what we are investigating (cancerous invasion vs tubulogenesis of non-transformed cells), and, importantly, the molecular mechanisms they have observed are different from what we report in our manuscript. They claimed a DDR1-RhoE- ROCK axis supports collective cell migration of epithelial cancer cells, in a manner independent from DDR1 kinase activity. On the other hand, our data extensively demonstrate that DDR1 kinase activity is involved in epithelial polarity. Thus, our discovery is different from previous reports and highlights importance in understanding epithelial polarity, a fundamental and essential process to maintain the lives of for multicellular organisms.

Reviewer #3

Overall evaluation:

The finding described here can be summarized as: (1) inhibition of the collagen-activated RTK DDR1 interferes with lumen formation in 3D but not 2D MDCK cell cultures and does so also in a mammary epithelia-derived cell line and in mammary organoids. (2) pharmacological RhoK/MyoII inhibition can overcome this defect. The authors complement these findings by showing that previously characterized defects in mammary duct development in DDR1-KO mice are associated with disrupted collagenIV/laminin deposition similar to their observation in MDCK cells.

No mechanism is forwarded, however. DDR1 is known, on the one hand, to co-localize and to interact with E-cadherin and on the other hand, to mediate Myosin II-dependent contraction at cell-ECM adhesion sites;

how DDR1 gets activated at the MDCK cell-cell junctions, how it signals there for RhoK/MyoII activation and how exactly RhoK/MyoII activity at cell-cell junctions regulates 3D morphogenesis was not investigated. That RhoA activity contributes to lumen formation in MDCK cells has been reported several times before and the mammary gland defect in DDR1 KO mice has also been previously described. In my opinion, at this state of knowledge a more mechanistic advance could be expected for a paper in this journal.

We agree that a clear mechanistic link is not defined by the data in our study. We thus discuss the possibility of a DDR1-SHP-2-Rho-ROCK axis as a potential mechanism in the discussion section.

Specific suggestions/comments:

I believe data presentation could be improved:

The authors start out analyzing HGF-mediated tubulogenesis, the most complex of the MDCK polarization assays, when the lumen defect is apparent in the cysts that form in simple 3D cultures. They also don't properly introduce their tubulogenesis assay and the distinct signaling steps it is known to involve, talking instead about "stimulation with morphogen".

Since the role of DDR1 in establishment of polarity emerged as we addressed its role in spatial regulation of MT1-MMP, we have decided to present the data in this order. According to the reviewer's comment, we replaced "morphogen" with HGF throughout.

Figs 2 and 3 are largely redundant, with Fig. 3 being far more informative. The MT1-MMP data do not connect to the rest of the findings. It remains unclear if the apical-to-basal MMP redistribution in 2D collagen cultures bears any significance for the polarity phenotype in the cysts.

We have merged Figures 2 and 3. The flow of the manuscript has improved by modifying the text.

In my opinion, the Figure legends are too detailed in the methodology.

Figure legends have been modified to shorten the length.

Additional comments:

The references Meder et al. and Torkko et al., are not the original reports that introduced Gp135 as MDCK cell apical marker and the cilium as a differentiation marker, respectively.

As the reviewer suggested, we now cite the original articles for GP135 (Ojakian, G.K., and R. Schwimmer. 1988. The polarized distribution of an apical cell surface glycoprotein is maintained by interactions with the cytoskeleton of Madin-Darby canine kidney cells. *J Cell Biol.* 107:2377-2387) and primary cilium (Vieira, O.V. et al. 2006. FAPP2, cilium formation, and compartmentalization of the apical membrane in polarized Madin-Darby canine kidney (MDCK) cells. *Proc Natl Acad Sci U S A.* 103:18556-18561).

Fig.S1: what are the multiple bands and arrowheads pointing to them?

We now modified the annotation in the Figure S1.

Fig. 5A: the E-cadherin staining is poor

E-cadherin in Figure 4A may not be very clear, but these are representative images. The quality of the data is also sufficient enough to show that E-cadherin is localised at lateral side of the membrane regardless of DDR1-IN-1 treatment. This is the same antibody used in Figure 4F which is also used by many other investigators and thus the staining is specific. The point we wish to make from this figure is that DDR1-IN-1 treatment does not influence apparent polarity under 2D culture conditions, and the the data is sufficient to support this notion.

Fig.6B needs some ppMLC quantitation because there are stronger and weaker labeled junctions apparent in both conditions.

Quantification was provided in Figure 6A, and Figure 6B provides qualitative data, showing that ppMLC

signal accumulates at cell-cell junctions co-localizing with ZO-1 upon DDR1-IN-1 treatment. Also, the original images were overexposed, thus the exposure has now been corrected to show proper staining.

In the x-z view in Fig.6C it appears that ctrl and DDR1-inhibited cells differ primarily in apical rather than junctional pMLC. The apical surface of polarized MDCK cells is not usually under myosin tension. Please explain what the apical contractile structures are. How was the specificity of the ppMLC staining controlled?

The same anti-ppMLC antibody has been widely used by many investigators and our data show that treating cells with Y27632 effectively abolishes the staining (new Figure 4C and Figure S7A). Thus, we trust the signal is specific. As the reviewer pointed out, increased ppMLC signals detected under DDR1-IN-1 treatment, but not in control cells, accumulate apical surface rather than lateral side(Figure 4D and E). This suggests that DDR1 inhibition leads to loss of the normal polarity of myosin activity, which our conclusion reflect. Double phosphorylated myosin is, as the reviewer states, not found on the apical domain, which is enriched in single phosphorylated myosin (Zihni et al NCB, 2017).

In Fig.6B a lot of the labeling appears to be diffusely intracellular and not obviously associated with fibers.

The images were overexposed. We have replaced these images with better exposed ones (new Figure 4D). As a result, ppMLC staining is cleaner.

February 7, 2019

RE: Life Science Alliance Manuscript #LSA-2018-00276-TR

Dr. Yoshifumi Itoh
University of Oxford
Kennedy Institute of Rheumatology
Roosevelt Drive
Headington
Oxford, Oxon OX3 7FY
United Kingdom

Dear Dr. Itoh,

Thank you for submitting your revised manuscript entitled "Epithelial polarization in 3D matrix requires DDR1 signaling to regulate actomyosin contractility". I appreciate the introduced changes and would be happy to publish your paper in Life Science Alliance pending final revisions necessary to meet our formatting guidelines:

- please add in the figure legends which statistical test has been used
- please note that figure panel 2G is currently not mentioned in the text (it seems there is an erroneously mentioning of Fig 3G instead)
- please mention Fig S7A in the legend

A. FINAL FILES:

-- High-resolution figure, supplementary figure and video files uploaded as individual files: See our detailed guidelines for preparing your production-ready images, <http://life-science-alliance.org/authorguide>

-- Summary blurb (enter in submission system): A short text summarizing in a single sentence the study (max. 200 characters including spaces). This text is used in conjunction with the titles of

papers, hence should be informative and complementary to the title. It should describe the context and significance of the findings for a general readership; it should be written in the present tense and refer to the work in the third person. Author names should not be mentioned.

B. MANUSCRIPT ORGANIZATION AND FORMATTING:

Full guidelines are available on our Instructions for Authors page, <http://life-science-alliance.org/authorguide>

Sincerely,

Andrea Leibfried, PhD
Executive Editor
Life Science Alliance
Meyershofstr. 1
69117 Heidelberg, Germany
t +49 6221 8891 502
e a.leibfried@life-science-alliance.org
www.life-science-alliance.org

February 8, 2019

RE: Life Science Alliance Manuscript #LSA-2018-00276-TRR

Dr. Yoshifumi Itoh
University of Oxford
Kennedy Institute of Rheumatology
Roosevelt Drive
Headington
Oxford, Oxon OX3 7FY
United Kingdom

Dear Dr. Itoh,

Thank you for submitting your Research Article entitled "Epithelial polarization in 3D matrix requires DDR1 signaling to regulate actomyosin contractility". It is a pleasure to let you know that your manuscript is now accepted for publication in Life Science Alliance. Congratulations on this interesting work.

*****IMPORTANT:** If you will be unreachable at any time, please provide us with the email address of an alternate author. Failure to respond to routine queries may lead to unavoidable delays in publication.*******

DISTRIBUTION OF MATERIALS:

Again, congratulations on a very nice paper. I hope you found the review process to be constructive and are pleased with how the manuscript was handled editorially. We look forward to future exciting

submissions from your lab.

Sincerely,
